# SHIFTING THE PARADIGM: A DIFFEOMORPHISM BETWEEN TIME SERIES DATA MANIFOLDS FOR ACHIEVING SHIFT-INVARIANCY IN DEEP LEARNING

**Berken Utku Demirel**
Department of Computer Science
ETH Zurich

**Christian Holz**
Department of Computer Science
ETH Zurich

## ABSTRACT

Deep learning models lack shift invariance, making them sensitive to input shifts that cause changes in output. While recent techniques seek to address this for images, our findings show that these approaches fail to provide shift-invariance in time series, where the data generation mechanism is more challenging due to the interaction of low and high frequencies. Worse, they also decrease performance across several tasks. In this paper, we propose a novel differentiable bijective function that maps samples from their high-dimensional data manifold to another manifold of the same dimension, without any dimensional reduction. Our approach guarantees that samples—when subjected to random shifts—are mapped to a unique point in the manifold while preserving all task-relevant information without loss. We theoretically and empirically demonstrate that the proposed transformation guarantees shift-invariance in deep learning models without imposing any limits to the shift. Our experiments on six time series tasks with state-of-the-art methods show that our approach consistently improves the performance while enabling models to achieve complete shift-invariance without modifying or imposing restrictions on the model's topology. The source code is available on GitHub.

## 1 INTRODUCTION

Inference on time series is essential for several important applications, such as heart rate (HR) estimation (Koshy et al., 2018), activity recognition (Saint-Maurice et al., 2020), and cardiovascular health monitoring (Hannun et al., 2019), which are generally performed using signals that are encoded as a sequence of discrete values over time. Most of these signals contain features that characterize the signal independently of their position in time (Waibel et al., 1989; Demirel & Holz, 2023). In other words, the information content of signals generally remains unchanged under the action of finite groups such as translations (Mallat, 2012). Therefore, ensuring the ability to accurately capture these inherent patterns is crucial for the reliability of the deep learning models in such critical human-involved health-related tasks (Akbar et al., 2019; Cutillo et al., 2020).

Deep learning networks perform downsampling by using strided-convolution and pooling (He et al., 2015; Krizhevsky et al., 2012), which cause loss of information due to high-frequency components of the input alias into lower frequencies, i.e., aliasing (Oppenheim et al., 1996). Previous works have proposed to employ a low-pass filter to prevent the aliasing and mitigate information loss during downsampling (Zhang, 2019; Mairal et al., 2014). While this additional filtering improved the robustness, the effect of employed low-pass filters is quite poor compared to the ideal implementation (see Figure 1 **a** and **b**), which still causes high-frequency components to alias into lower ones.

Despite the potential benefits of emphasizing low-frequency components for image recognition, as it aligns with human perception (Subramanian et al., 2023), the imperfect preservation of frequency components with each subsampling layer contributes to information loss, leads to performance degradation, especially in tasks where the significance lies in both low and high-frequency components with their interactions. A more recent approach to achieve shift-invariant neural networks involves the use of adaptive subsampling grids (Chaman & Dokmanic, 2021). However, these methods still

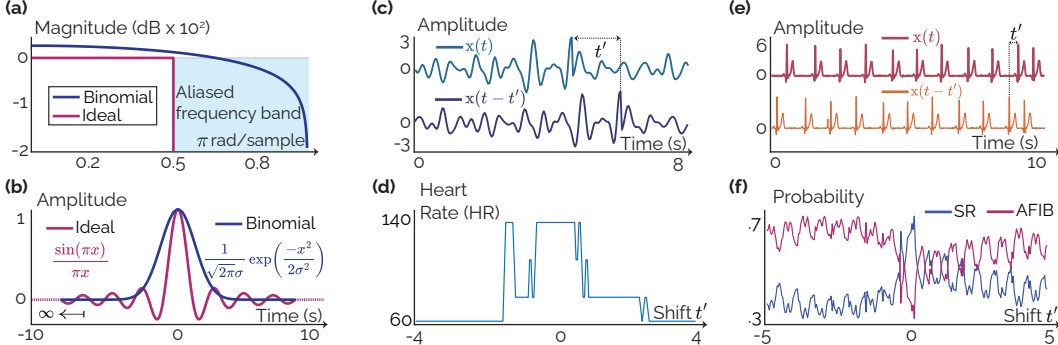

Figure 1: **(a)** The magnitude response of the ideal low-pass filter and binomial filter that is employed in (Zhang, 2019) for preventing aliasing. **(b)** Time domain representations of the ideal and binomial filters with interpolation for smoother waveforms. **(c)** An 8-second signal for blood volume changes and its $t'$ shifted version, obtained through photoplethysmogram——a widely utilized signal for heart rate monitoring (Perez et al., 2019). **(d)** The heart rate prediction of a trained ResNet with binomial filters to prevent aliasing. Different amounts of shifts ($t' \in [-4, 4]$) change the trained model output drastically from 140 to 60 beats per minute (bpm). **(e)** A 10-second electrocardiogram (ECG) signal from a patient with atrial fibrillation (AFIB). **(f)** The model misclassifies the abnormal AFIB pattern as a healthy sinus rhythm (SR), with shifts causing a complete change in output probability.

fail to guarantee shift-invariance due to the change in content at the boundary (Rojas-Gomez et al., 2022) and impose constraints on the shift range to maintain invariance.

Consequently, the evaluation of these methods is confined to a limited range of shifts while covering a small subset of the space. Additionally, their reliance on a grid scheme introduces a dependence on sampling rates, resulting in performance gaps across the entire shift space (Michaeli et al., 2023).

In this work, we propose a differentiable bijective function that maps samples from their high-dimensional data manifold to another manifold of the same dimension, without any dimensional reduction. Our method ensures that randomly shifted samples—representing variations of the same signal—are mapped to the same point in the space, preserving all task-relevant information.

Since our method modifies the data space, it can be integrated into any deep learning architecture, offering an adaptable and complementary solution for achieving shift-invariance in time series. Summarizing our contributions in this paper:

- We introduce a novel diffeomorphism to ensure shift-invariance in neural networks. Additionally, we incorporate the proposed diffeomorphism into the network architecture using a novel, tailored loss term to further enhance performance while ensuring invariance.

- We demonstrate both theoretically and empirically that the proposed transformation guarantees shift-invariance in models without imposing any limits to the range of shifts or changing model topology, which enable previous methods to be used in conjuction.

- We conduct extensive experiments on six time series tasks with nine datasets. Our experiments show that the proposed approach consistently improves the performance while decreasing the variance and enabling models to achieve complete shift-invariance.

## 2 METHOD

### 2.1 NOTATIONS

We use bold lowercase symbols ($\mathbf{x}$) for time series. The parametric mappings are represented as $f_\theta(.)$ where $\theta$ is the parameter. The discrete Fourier transformation of a time series is denoted as $\mathcal{F}(\mathbf{x})$, yielding a complex variable $|X(e^{j\omega})|e^{j\phi(\omega)}$ which contains magnitude and phase information of each harmonic (sinusoidal). $\phi(\omega_k)$ and $T_k$ represent the phase angle and period of the $k$-th harmonic with frequency $\omega_k$. We mainly used the textbook notations (Oppenheim et al., 1996) throughout the script, providing a comprehensive list of notations and detailed definitions in Appendix A.1.

## 2.2 OBJECTIVE

Given a dataset $\mathcal{D} = \{(\mathbf{x}(t)_i, \mathbf{y}_i)\}_{i=1}^{K}$ where each $\mathbf{x} \in X$ consists of uniformly sampled real-valued values and each $\mathbf{y} \in Y$ represents the corresponding labels, the objective is to have consistent and accurate outputs for all variants of a sample that are subjected to shifts[1] such that when a parametric model $f_\theta : X \to Y$ is evaluated on the set $\mathcal{D}_{\text{test}} = \{(\mathbf{x}(t)_i, \mathbf{y}_i)\}_{i=1}^{L}$, the output will be the same and true $\mathbf{y}_i$ for all $t'$ to be shift-invariant, i.e., $\mathbf{y}_i = f_\theta(\mathbf{x}(t - t')_i), \forall t' \in \mathbb{R}$.

We propose a diffeomorphism that maps randomly shifted time series samples to the same point in data space, preserving all relevant information to ensure shift-invariance. The motivation and theoretical derivation of our method are presented in the following steps.

**Proposition 2.1** (Time shift as a Group Operation). *Shift operation in time domain defines an Abelian Group of phase angles in the frequency domain for each harmonic with frequency $\omega_k$.*

$$(\Phi_k, + \mod 2\pi), \quad where \quad \Phi_k = \{\phi \mid \phi = (\phi(\omega_k) + \omega_k t') \mod 2\pi, t' \in \mathbb{R}\} \tag{1}$$

*Proof.* Using $\mathcal{F}(x(t + t')) = |X(e^{j\omega})|e^{j\phi(\omega)}e^{j\omega t'}$, and the multiplication of complex numbers

$$\exists t' \in \mathbb{R}, \ \forall \phi \in (-\pi, \pi], \ \ \phi = (\phi(\omega_k) + \omega_k t') \mod 2\pi \tag{2}$$

See Appendix A for detailed proof with group axioms. $\qquad\square$

Proposition 2.1 states that the shift variants of a sequence define a group of phase angles, known as circle group (Fuchs, 1960) $\mathbb{T}$. An important observation from Equation 2 is that different shift values ($t'$) can map to the same phase angle ($\phi$) due to modulo operation with $2\pi$.

However, a closer look reveals that this mapping can be defined uniquely for specific harmonics using the circular shift. Specifically, we can represent every point in the shift space uniquely with the phase angle of a harmonic whose period is equal to or longer than the length of sample, i.e., $T_0 \le t$. In the remainder of this section, we explain how this observation is framed as a novel diffeomorphism. We denote the frequency, period, and phase of this specific harmonic as $\omega_0$, $T_0$, and $\phi(\omega_0)$, respectively.

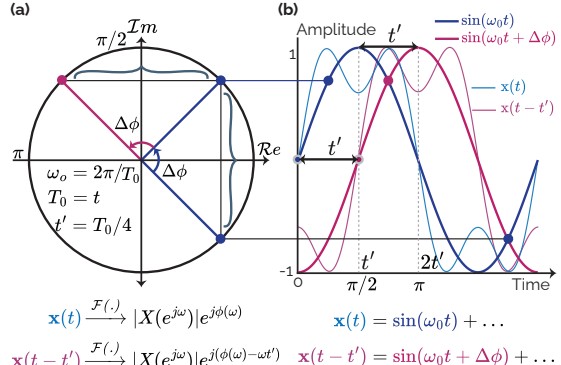

$$\mathbf{x}(t) \xrightarrow{\mathcal{F}(\cdot)} |X(e^{j\omega})|e^{j\phi(\omega)} \qquad \mathbf{x}(t) = \sin(\omega_0 t) + \dots$$
$$\mathbf{x}(t - t') \xrightarrow{\mathcal{F}(\cdot)} |X(e^{j\omega})|e^{j(\phi(\omega) - \omega t')} \quad \mathbf{x}(t - t') = \sin(\omega_0 t + \Delta\phi) + \dots$$

Figure 2: **(a)** Frequency domain representation of a harmonic at frequency $\omega_0$ with different phase angles in unit circle. **(b)** Time domain representation of a signal $\mathbf{x}(t)$ and its shifted version $\mathbf{x}(t - t')$. The phase angle of the harmonic can cover all (i.e., surjective $\mathcal{T}(\mathbf{x}, \phi)$) potential shifts. Moreover, shifts in the time domain correspond to unique (i.e., injective $\mathcal{T}(\mathbf{x}, \phi)$) angle rotations in the frequency domain for the sinusoidal with periodicity $T_0$. Therefore, the proposed transformation function $\mathcal{T}(\mathbf{x}, \phi)$ is bijective.

The proposed transformation function, $\mathcal{T}(\mathbf{x}, \phi)$, takes a sample $\mathbf{x}$ and an angle $\phi \in (-\pi, \pi]$. It then applies a linear phase shift to each harmonic, mapping the time series to a new variant where the phase angle of the harmonic with frequency $\omega_0$ matches the desired angle $\phi$. The proposed transformation, which converts a time series to another shifted variant, is defined as in Equations 3 and 4.

$$\mathbf{x}(t) \xrightarrow{\mathcal{T}(\mathbf{x}, \phi)} \mathcal{F}^{-1}(|X(e^{j\omega})|e^{j\phi(\omega)}e^{-j\omega\Delta\phi}) \quad \text{where} \tag{3}$$

$$\Delta\phi = \begin{cases} \frac{(\theta - 2\pi) * T_0}{2\pi}, & \text{if } \theta > \pi \\ \frac{\theta * T_0}{2\pi}, & \text{else} \end{cases} \quad \text{and} \quad \theta = [\phi(\omega_0) - \phi] \% 2\pi \tag{4}$$

Mainly, the transformation first decomposes a signal to its harmonics, then it calculates the phase difference, denoted as $\Delta\phi$, between the harmonic with frequency $\omega_0$ and the desired angle $\phi$. Finally,

---

[1]We represent a time shift ($t'$) for a sample $\mathbf{x}$ as $\mathbf{x}(t - t')$, similar to Oppenheim et al. (1996). All the time shifts throughout the paper imply circular shift, i.e., $(t - t') = (t - t')_{\%t}$ where $\%$ is the modulus.

It returns to the time domain by taking the inverse Fourier transform, $\mathcal{F}^{-1}(.)$, while applying a linear phase shift to all harmonics to preserve the waveform morphology. In the end, the transformation matches the phase angle of the harmonic at frequency $\omega_0$ with the desired angle $\phi$. We first demonstrate that the proposed transformation is a bijective function, as shown in Theorem 2.2.

**Theorem 2.2** (Covering the Entire Time Space Injectively). *Given a sample* $\mathbf{x}$*, the defined function* $\mathcal{T}(\mathbf{x}, \phi) : \Phi \times \mathbb{R}^d \to \mathbb{R}^d \times \Delta\Phi$ *is bijective such that all shift variants of a sample can be covered with the unique phase angle of a harmonic whose period is longer or equal to the length of* $\mathbf{x}$*.*

$$\forall \phi_a, \phi_b \in \Phi, \ \mathcal{T}(\mathbf{x}, \phi_a) = \mathcal{T}(\mathbf{x}, \phi_b) \implies \phi_a = \phi_b$$
$$\forall t' \in \mathbb{R}, \ \exists \phi \in \Phi, \ \mathcal{T}(\mathbf{x}, \phi) = (\mathbf{x}(t - t'), \Delta\phi),$$

*where the first and second equations represent the injection and surjection, respectively.*

We provide an intuitive demonstration in Figure 2, with a detailed mathematical proof in Appendix A. Since each point in the shift space can be uniquely defined by the phase angle of a harmonic with period $T_0$, we use the angle of this harmonic to define manifolds [2], $\mathcal{M}^\phi$, on which the samples lie. Specifically, we apply the proposed transformation $\mathcal{T}(\mathbf{x}, \phi)$ for each sample to map it to a manifold defined by the angle, i.e., $\mathcal{T}(\mathbf{x}, \phi_a) \in \mathcal{M}^{\phi_a}$, $\mathcal{T}(\mathbf{x}, \phi_b) \in \mathcal{M}^{\phi_b}$, and $\bigcap_{i=0}^{2\pi} \mathcal{M}^{\phi_i} = \emptyset$ (See Appendices A.1.2 and A.2 for detailed definition of manifolds and notations). We, therefore, can map a sample and its randomly shifted variants to the same point in the space, which is sufficient for providing shift-invariance as demonstrated in Theorem 2.3 with a detailed proof in Appendix A.

**Theorem 2.3** (Guarantees for Shift-Invariance). *Given* $\mathbf{x}$ *and a randomly shifted variant of it* $\mathbf{x}(t-t')$*, if* $\mathcal{T}(\mathbf{x}, \phi)$ *is applied to both samples with the same angle* $\phi_a$*, the resulting samples will be the same.*

$$\mathcal{T}(\mathbf{x}(t), \phi_a) = \left(\tilde{\mathbf{x}}(t), \ \Delta\phi_{\mathbf{x}(t)}\right), \quad \mathcal{T}(\mathbf{x}(t - t'), \phi_a) = \left(\tilde{\mathbf{x}}(t), \ \Delta\phi_{\mathbf{x}(t-t')}\right)$$

*Proof.*

$$\phi_{\mathbf{x}(t)} = \phi(\omega), \ \phi_{\mathbf{x}(t-t')} = \phi(\omega) - \omega t', \quad \Delta\phi_{\mathbf{x}(t-t')} - \Delta\phi_{\mathbf{x}(t)} = -\omega_0 \frac{T_0}{2\pi} t' \quad (5)$$

$$\phi_{\mathcal{T}(\mathbf{x}(t-t'), \phi_a)} - \phi_{\mathcal{T}(\mathbf{x}, \phi_a)} = \left[\frac{T_0}{T}\omega_0 - \omega\right] t', \quad \phi_{\mathcal{T}(\mathbf{x}, \phi_a)} = \phi_{\mathcal{T}(\mathbf{x}(t-t'), \phi_a)} \quad (6)$$

Therefore, the output time series samples will be the same after applying the transformation. $\square$

The proof concludes by demonstrating that the harmonics retain the same phase and magnitude after transformation, despite an unknown shift applied to the sample. Moreover, since the transformation only contains exponentials with Fourier transform, it is fully differentiable, allowing optimization with

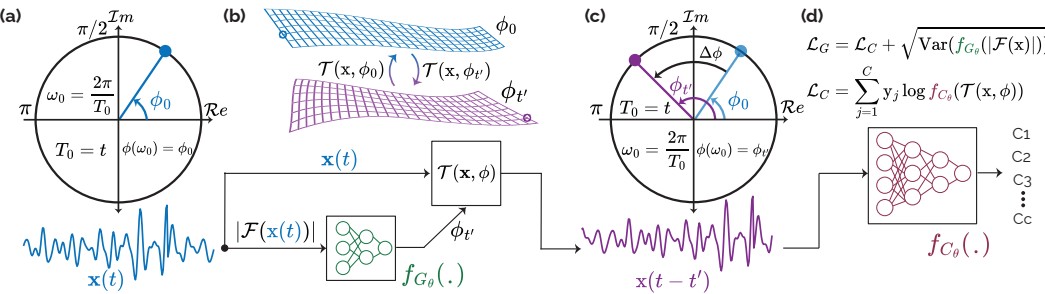

Figure 3: **(a)** An input signal in the time domain and complex plane representation of its decomposed sinusoidal of frequency $\omega_0 = \frac{2\pi}{T_0}$ with the phase angle $\phi_0$. **(b)** Guiding the diffeomorphism to map samples between manifolds. **(c)** The obtained waveform with a phase shift applied to all frequencies linearly, calculated by the angle difference, as in Equation 4, without altering the waveform. **(d)** The loss functions for optimizing networks with the cross-entropy and the variance of possible manifolds.

---

[2]The manifold is defined as a $d$-dimensional Euclidean space, matching the data's dimension, to better explain the abstract transformation. There is no manifold learning of low-dimensional space in our transformation.

neural networks. Therefore, we use a guidance network $f_{G_\theta} : \mathbb{R}^d \to \Phi$ with a shift-invariant input, absolute Fourier transform of samples, to generate an angle in radians for mapping. Simultaneously, the main classifier $f_{C_\theta} : X \to Y$ maps the transformed samples to the label space. Both networks are optimized with cross-entropy loss. The optimizer for the guidance network ($\mathcal{L}_G$) has an additional loss term to reduce variations in a batch ($\mathcal{B}$) of angles, as given in Equation 7.

$$\mathcal{L}_C = -\sum_{i=1}^N \sum_{j=1}^C y_{ij} \log f_{C_j}(\mathcal{T}(\mathbf{x}_i, \phi_i)) \quad \mathcal{L}_G = \mathcal{L}_C + \sqrt{\mathrm{Var}_{\mathbf{x} \sim \mathcal{B}}\left(f_{\theta_G}\left(|\mathcal{F}(\mathbf{x})|\right)\right)} \quad (7)$$

The guidance network, optimized by the proposed loss, works as an adaptive linear constraint that limits the regions in the original data space where samples can be found. In other words, if we conceptualize the data space of samples as expanding with shift variants, as illustrated in Figure 3, the model learns to reduce the potential points where samples can be found in the data space.

Additionally, for real-world samples where optimal phase shift values are unavailable, applying trivial phase shifting may lead to suboptimal data space representations. To address this, we transform the data space using the proposed diffeomorphism for the downstream tasks using the guidance network (see Appendix F for a detailed analysis of the guidance network). Moreover, unlike traditional manifold learning methods (Lin & Zha, 2008; Wang et al., 2004), which project data into lower-dimensional spaces, our approach operates directly within the original data space. In our ablation studies, we thoroughly examine the impact of loss terms on the performance and present the findings.

## 3 EXPERIMENTS

### 3.1 DATASETS

We conducted experiments on nine datasets across six tasks, including heart rate (HR) estimation from photoplethysmography (PPG), step counting and activity recognition using inertial measurements (IMUs), cardiovascular disease classification from electrocardiogram (ECG), sleep stage classification from electroencephalography (EEG) and lung sound classification from audio. We provide short descriptions of each dataset below, and further details can be found in Appendix C.

***Heart rate***    We used the IEEE Signal Processing Cup in 2015 (IEEE SPC) (Zhang et al., 2015), and DaLia (Reiss et al., 2019) for PPG-based heart rate prediction. We used the leave-one-session-out (LOSO) cross-validation, which evaluates models on subjects/sessions that were not used for training.

***Activity recognition***    We used UCIHAR (Anguita et al., 2012), and HHAR (Stisen et al., 2015) for activity recognition from inertial measurement units from smartphones. We evaluate the cross-person generalization performance of the models, i.e., the model is evaluated on previously unseen subjects.

***Cardiovascular disease (CVD) classification***    We used Chapman University, Shaoxing People's Hospital ECG (Zheng et al., 2020) and PhysioNet 2017 (Clifford et al., 2017; Goldberger et al., 2000) datasets. We selected the same four leads for the Chapman as in (Alday et al., 2020). We split the datasets into training, validation, and test sets according to the patient ID (each patient's recordings appear in only one set) using a 60, 20, 20 ratio as in Demirel & Holz (2023); Zheng et al. (2020).

**Step counting**    We used the Clemson dataset (Mattfeld et al., 2017), which released for pedometer evaluation. We conducted experiments using wrist IMUs where labels are available through videos.

***Sleep stage classification***    We used the Sleep-EDF dataset, from PhysioBank (Goldberger et al., 2000), which includes whole-night PSG sleep recordings, where we used a single EEG channel (i.e., Fpz-Cz) with a sampling rate of $100\,\mathrm{Hz}$, following the same setup as in Eldele et al. (2021).

***Lung sound classification***    We used the Respiratory@TR, which contains lung sounds recorded with two digital stethoscopes (Altan et al., 2017). Two pulmonologists validated and labeled the recordings based on X-rays, pulmonary function tests (PFTs), and auscultation. The labels correspond to five COPD severity levels (COPD0–COPD4) as described in prior work (Zhang et al., 2024).

## 3.2 BASELINES

We compared our method and existing approaches including low-pass filtering (LPF) (Zhang, 2019), and adaptive subsampling grids (APS) (Chaman & Dokmanic, 2021). In addition to shift-invariance techniques, we evaluated our method against shift-equivariant Wavelet Networks (Romero et al., 2024) and canonical representation learning techniques for equivariance (Kaba et al., 2023; Mondal et al., 2023). Moreover, since our method can be integrated with any existing approaches, we investigate the performance of previous techniques for shift-invariancy when combined with our algorithm.

## 3.3 IMPLEMENTATION

We follow a similar implementation setup as previous work on shift-invariancy (Zhang, 2019) in supervised learning, making architectural adjustments for time series. Specifically, we employed ResNet (He et al., 2015) with eight blocks designed for time series (Hong et al., 2020), excluding signals from inertial measurement units with a single dimension. For the latter, we observed a better performance with fully connected networks (FCN). Therefore, we used a three-layer FCN for the single dimensional IMU-based task, i.e., step counting. Similarly, for guiding the transformation function, we used an FCN with a single output, which is the angle for the chosen sinusoidal. For each dataset, we set the Fourier transform length equal to the signal length, as the Fourier transformation of the same size inherently includes sinusoids with periods equal to or longer than the signal length. We use categorical cross-entropy loss, which is optimized using Adam (Kingma & Ba, 2015). The learning rate is determined through grid search for each dataset and set to the same value for all baselines given in the Appendix. During training, it was halved when the validation loss stops improving for 15 consecutive epochs. The training is terminated when 90 successive epochs show no validation performance improvements. The best model is chosen as the lowest loss on the validation set. Detailed hyperparameters and architecture specifications can be found in the Appendix C.4.

## 3.4 EVALUATION

We evaluate the performance of the models using the common evaluation metrics, i.e., accuracy, F1, for each task. For shift-invariancy, we used the shift consistency (S-Cons.) metric which measures how often the network outputs the same classification, given the same time series with two different shifts, similar to (Zhang, 2019) as in Equation 8. We applied shifts across the entire space in contrast to previous approaches where the range of shift is heavily limited (Rojas-Gomez et al., 2022).

$$\mathbb{E}_{X,t_1,t_2} \mathbb{1}\left[ \hat{f}_C(\mathrm{x}(t-t_1)) = \hat{f}_C(\mathrm{x}(t-t_2)) \right], \tag{8}$$

where $\hat{f}_C$ represents the classifier's output following the arg max operation. $t_{1,2}$ are uniformly sampled integers from the interval $[1, t]$, with $t$ denoting the length of the sample.

## 4 RESULTS AND DISCUSSION

We present the main results of our approach compared to state-of-the-art methods across the six time series tasks on nine datasets. Overall, our method has demonstrated a substantial performance improvement, reaching up to 10–15% in some tasks, while increasing the shift consistency up to 50–60% compared to previous techniques.

The experimental results from all the time series tasks are given in Tables 1, 2, 3 and 4. These tables demonstrate that the previous techniques fail to provide shift-invariant models when applied to time series without limiting shifts. Additionally, the models exhibit extremely low consistency (as low as 32%) in HR prediction. More importantly, applying state-of-the-art methods to enhance shift consistency in deep learning models for predicting the heart rate results in performance degradation.

We believe the main reason for the small improvements in the consistency of previous techniques is that the research to date has tended to focus on limited shifts rather than considering the whole shift space as literature is mostly concerned about images. While restricting shifts can be a valid assumption in computer vision, where the main reasoning is that the object being classified should not be near the boundary. This assumption does not apply to time series, where the whole signal

Table 1: Performance comparison of our method and other techniques for HR estimation

| Method | IEEE SPC22 | | | | DaLiA | | | |
|---|---|---|---|---|---|---|---|---|
| | S-Cons (%) ↑ | RMSE ↓ | MAE ↓ | $\rho$ (%) ↑ | S-Cons (%) ↑ | RMSE ↓ | MAE ↓ | $\rho$ (%) ↑ |
| Baseline | 61.99±1.19 | 18.39±2.96 | 10.28±1.41 | 62.64±5.74 | 32.08±0.22 | 9.86±0.23 | 4.40±0.03 | 86.01±0.51 |
| Aug. | 76.48±1.77 | 18.73±1.15 | 10.42±0.40 | 64.06±3.70 | 52.77±0.39 | 9.85±0.21 | 4.47±0.06 | 85.99±0.49 |
| LPF | 76.88±0.73 | 20.20±1.54 | 13.44±0.82 | 65.40±1.92 | 38.67±0.30 | 10.01±0.30 | 4.67±0.12 | 85.68±0.51 |
| APS | 73.99±1.06 | 19.42±0.60 | 12.98±0.29 | 65.27±1.32 | 44.33±0.16 | 10.45±0.40 | 5.01±0.17 | 84.69±0.85 |
| WaveletNet | 51.71±1.95 | 21.56±1.01 | 14.61±0.34 | 60.74±4.37 | 36.71±3.04 | 15.46±0.64 | 7.67±0.23 | 76.13±1.86 |
| Canonicalize | 63.52±1.20 | 19.02±0.62 | 10.40±0.69 | 61.27±1.07 | 32.01±0.33 | 9.77±0.12 | 4.39±0.05 | 86.02±0.30 |
| Ours | **100±0.00** | **16.25±0.72** | **9.45±0.03** | **70.12±2.10** | **100±0.00** | **9.75±0.15** | **4.39±0.03** | **86.06±0.19** |
| Ours+LPF | 100±0.00 | 20.34±1.62 | 13.77±0.84 | 65.60±2.31 | 100±0.00 | 10.72±0.11 | 5.30±0.03 | 84.12±0.23 |
| Ours+APS | 100±0.00 | 18.81±1.59 | 12.32±0.84 | 67.01±3.79 | 100±0.00 | 10.47±0.09 | 5.10±0.03 | 84.62±0.31 |

Table 2: Performance comparison of ours and other techniques in *ECG* datasets for CVD classification

| Method | Chapman | | | | PhysioNet 2017 | | | |
|---|---|---|---|---|---|---|---|---|
| | S-Cons (%) ↑ | Acc ↑ | F1 ↑ | AUC (%)↑ | S-Cons (%) ↑ | Acc ↑ | F1 ↑ | AUC ↑ |
| Baseline | 98.53±0.17 | 91.32±0.23 | 91.22±0.24 | 98.34±0.16 | 98.37±0.15 | 83.22±0.72 | 73.50±1.99 | 93.21±0.30 |
| Aug. | 99.00±0.16 | 91.96±0.19 | 91.89±0.22 | 98.45±0.18 | 98.96±0.17 | 82.28±1.18 | 72.32±2.20 | 93.20±0.42 |
| LPF | 98.69±0.14 | 92.01±0.23 | 91.94±0.58 | 98.50±0.24 | 98.94±0.39 | 84.40±0.16 | 75.68±0.76 | 93.80±0.32 |
| APS | 98.60±0.17 | 90.69±0.89 | 89.44±1.00 | 98.31±0.24 | — | — | — | — |
| WaveletNet | 91.02±1.14 | 90.87±1.02 | 90.02±1.00 | 97.94±0.21 | 65.03±0.71 | 76.06±0.64 | 63.35±3.40 | 87.02±0.29 |
| Canonicalize | 98.80±0.24 | 91.93±0.13 | 90.87±0.18 | 98.42±0.15 | 98.26±0.31 | 83.34±0.46 | 73.97±0.67 | 93.68±0.31 |
| Ours | **100±0.00** | **92.10±0.25** | 91.93±0.85 | 98.47±0.15 | **100±0.00** | 83.15±0.65 | 74.12±1.80 | 93.28±0.31 |
| Ours+LPF | 100±0.00 | 92.05±0.52 | **91.96±0.54** | **98.51±0.10** | 100±0.00 | **85.20±0.40** | **77.50±1.21** | **94.20±0.19** |
| Ours+APS | 100±0.00 | 91.61±1.11 | 91.10±0.56 | 98.36±0.20 | — | — | — | — |

carries the information (Demirel & Holz, 2023) additional to local waveform features, and as such, there is no explicit boundary condition or input area to consider for limiting the range of shifts.

The empirical results support our motivation for proposing a differentiable bijective function that maps samples with different shifts to the same point on the data manifold, avoiding the limited shift assumption. Additionally, applying low-pass filtering to prevent aliasing can degrade performance for certain tasks, where the interaction between frequencies plays a critical role (Canolty et al., 2006).

**Time delay as adversary?** An interesting finding from our experiments is the notable decline in model consistency as the number of output classes increases. This behavior in the models is similar to previous findings on adversarial examples, indicating that the robustness decreases with a higher number of classes (Fawzi et al., 2018). During our experiments, we observed the same phenomenon where the small shifts of the input change the output to another class, particularly when the task complexity increased with a higher number of classes. For example, in the case of HR estimation (Table 1), even short shifts (as low as 10–100 ms) can lead to a change in the prediction by over 80 bpm, despite no alteration in the periodicity of the signal, which is the main feature for this task.

Normally, it is expected that models learn the periodicities in these signals and infer the heart rate. However, our results indicate that the models learn something else or in a different way, because as the signal undergoes a slight shift, the model prediction jumps more than 100%, even though the periodicity of the waveform remains unchanged with the shift operation.

We believe these drastic output changes arise from the model's sensitivity to (shortcut) features (Geirhos et al., 2020; Zhang et al., 2021), resulting in a performance decrease when evaluated on samples different from those encountered during training. Since our proposed transformation

Table 3: Performance comparison of our method with other techniques on an *EEG* dataset for sleep stage classification and an *audio* dataset for lung sound classification in respiratory health assessment

| Method | Sleep-EDF | | | | Respiratory | | | |
|---|---|---|---|---|---|---|---|---|
| | S-Cons (%) ↑ | Acc ↑ | W-F1 ↑ | $\kappa$ ↑ | S-Cons (%) ↑ | Acc ↑ | F1 ↑ | W-F1 ↑ |
| Baseline | 95.06±0.61 | 75.41±2.01 | 74.87±1.92 | 67.12±2.96 | 99.10±0.43 | 25.21±5.60 | 57.01±3.62 | 21.21±5.98 |
| Aug. | 99.00±0.17 | 74.89±1.11 | 74.03±1.46 | 65.89±1.81 | 99.68±0.42 | 20.32±5.18 | 45.81±3.51 | 15.31±6.07 |
| LPF | 92.43±1.24 | 73.56±2.93 | 76.01±1.98 | 65.68±3.46 | 99.50±0.42 | 19.47±9.78 | 46.53±3.04 | 11.89±4.98 |
| WaveletNet | 84.40±5.90 | 73.54±4.78 | 72.74±3.45 | 64.66±4.12 | 91.38±2.40 | 28.57±10.81 | 44.23±7.12 | 17.10±7.81 |
| Canonicalize | 93.95±0.51 | 77.12±2.21 | 70.14±2.25 | 69.81±2.76 | 98.28±0.64 | 22.68±10.52 | 45.33±5.75 | 15.30±5.33 |
| Ours | **100±0.00** | **77.90±1.92** | **76.77±2.58** | **70.01±1.10** | **100±0.00** | **33.10±5.12** | **60.13±4.67** | **28.33±6.55** |
| Ours+LPF | 100±0.00 | 73.12±1.89 | 75.34±1.61 | 64.98±2.27 | 100±0.00 | 25.77±2.12 | 51.82±2.10 | 17.99±4.15 |

Table 4: Performance comparison of our method with others in *IMU* datasets for Activity and Step

| Method | UCIHAR | | | HHAR | | | Clemson | | |
|---|---|---|---|---|---|---|---|---|---|
| | S-Cons (%) ↑ | Acc ↑ | F1 ↑ | S-Cons (%) ↑ | Acc ↑ | F1 ↑ | S-Cons (%) ↑ | MAPE ↓ | MAE ↓ |
| Baseline | 94.07±1.38 | 85.39±2.30 | 83.20±2.94 | 98.27±0.33 | 91.87±1.36 | 91.16±1.38 | 54.31±4.40 | 4.76±0.11 | 2.74±0.08 |
| Aug. | 96.55±0.80 | 85.42±4.50 | 83.69±6.74 | 98.38±0.28 | 91.97±0.44 | 91.31±0.49 | 61.01±4.88 | 4.08±0.14 | 2.29±0.07 |
| LPF | 95.05±0.21 | 83.96±3.44 | 81.08±4.21 | 98.10±0.10 | 92.10±0.80 | 91.43±0.94 | 59.77±4.40 | 4.16±0.16 | 2.35±0.11 |
| APS | 96.40±0.03 | 81.75±4.11 | 79.01±5.33 | 98.30±0.24 | 91.83±1.35 | 91.01±1.47 | 45.50±2.69 | 4.74±0.16 | 2.69±0.07 |
| WaveletNet | 94.56±1.31 | 82.78±4.62 | 80.73±5.59 | 96.76±0.15 | 90.72±0.38 | 90.71±0.39 | 59.14±3.10 | 5.20±0.66 | 2.95±0.41 |
| Canonicalize | 97.72±0.37 | 84.10±2.10 | 81.89±2.89 | 98.27±0.07 | 91.56±1.18 | 90.73±1.10 | 55.47±4.87 | 4.54±0.46 | 2.59±0.29 |
| Ours | **100**±0.00 | **87.71**±1.98 | **85.67**±2.47 | **100**±0.00 | 91.93±1.14 | 91.12±1.03 | **100**±0.00 | 4.28±0.34 | 2.43±0.21 |
| Ours+LPF | **100**±0.00 | 84.78±2.46 | 82.58±2.62 | **100**±0.00 | **92.51**±0.55 | **91.80**±0.62 | **100**±0.00 | **3.75**±0.33 | **2.12**±0.18 |
| Ours+APS | **100**±0.00 | 82.96±1.79 | 81.10±1.73 | **100**±0.00 | 91.38±0.32 | 90.64±0.32 | **100**±0.00 | 3.87±0.19 | 2.19±0.11 |

function works as an adaptive linear constraint in the data space, it reduces the potential points where samples can exist, thereby enhancing overall performance.

One distinct result from our experiments is that when previous shift-invariancy techniques are applied to the heart rate prediction task, the average error rate of the models increases by 7–10%. This performance decrease can be easily observed in the DaLiA (Table 1) for the adaptive sampling technique. The performance discrepancy between tasks can be attributed to the dataset and signal characteristics. Since DaLiA contains impulse random noise with multiple periodicities, the norm-based subsampling can inadvertently emphasize the noisy waveforms instead of the desired pattern during the subsampling of feature maps, leading to a decrease in prediction performance.

We conduct detailed ablation experiments to further investigate the impact of various components, with a particular focus on the effect of the proposed mapping function under different modifications, i.e., modified loss for optimization, on the overall model's performance across time series tasks.

## 4.1 ABLATION STUDY

We present a comprehensive investigation of our method and the effect of its components on the performance. Mainly, we investigate the effect of guiding the proposed transformation with different loss functions and without any guidance. First, we map all samples to a single manifold $\mathcal{M}^{\phi_0}$, i.e., $\mathcal{T}(\mathbf{x}, \phi)$ is applied with a constant $\phi = 0$ instead of learning the angle for each sample. We experimented with different values of $\phi \sim (-\pi, \pi]$, but observed no significant change in the performance when the mapped manifold is constant for samples. Second, we modify the loss for training the guidance network to increase the variance of angles—increasing the possible manifolds where data can be found—without changing the cross-entropy loss from the classification network as in Equation 9, ($\hat{\mathcal{L}}_G$). Finally, we train both networks only with the cross-entropy loss ($\mathcal{L}'_G = \mathcal{L}_C$). We compared these three variants of the learning techniques with the original proposed implementation as each represents distinct approaches for manipulating the data space. For example, when all samples are mapped to a single manifold, the variations in samples decrease significantly since there is only one possible phase angle for the chosen harmonic with period $\mathrm{T}_0$. Additionally, the relationships among all sinusoidal components remain invariant, given that the proposed transformation is a linear function of the frequency. Conversely, optimizing the guidance network to increase the variance of angles, thereby favoring a greater sample diversity, expands the possible variations for samples.

$$\hat{\mathcal{L}}_G = \mathcal{L}_C - \sqrt{\mathrm{Var}_{\mathbf{x} \sim \mathcal{B}} \left( f_{\theta_G} \left( |\mathcal{F}(\mathbf{x})| \right) \right)} \tag{9}$$

Tables 5 and 6 summarize the results where we exclude the consistency metric from the tables as the models that include the proposed transformation are always completely shift-invariant. The first row ($\mathcal{T}(\mathbf{x}, \phi)$) in the tables shows the performance when all the samples are mapped to a single manifold

Table 5: Ablation experiments for *HR* (left) and *IMU* (right) tasks

| Method | IEEE SPC22 | | | DaLiA$_{PPG}$ | | | Method | UCIHAR | | HHAR | | Clemson | |
|---|---|---|---|---|---|---|---|---|---|---|---|---|---|
| | MAE ↓ | RMSE ↓ | $\rho$ ↑ | MAE ↓ | RMSE ↓ | $\rho$ ↑ | | Acc ↑ | F1 ↑ | Acc ↑ | F1 ↑ | MAPE ↓ | MAE ↓ |
| $\mathcal{T}(\mathbf{x}, \phi)$ | 11.15 | 19.18 | 62.07 | 4.77 | 10.13 | 85.35 | $\mathcal{T}(\mathbf{x}, \phi)$ | 84.67 | 82.65 | **92.33** | **91.56** | 4.64 | 2.67 |
| $\mathcal{L}'_G$ | 9.80 | 17.16 | 66.80 | 4.60 | 10.10 | 85.52 | $\mathcal{L}'_G$ | 84.30 | 82.49 | 91.98 | 91.18 | 4.42 | 2.52 |
| $\hat{\mathcal{L}}_G$ | 9.45 | 17.00 | 69.10 | 4.41 | **9.63** | **86.35** | $\hat{\mathcal{L}}_G$ | 84.82 | 81.99 | 91.51 | 90.83 | 4.31 | 2.45 |
| Ours | **9.45** | **16.25** | **70.12** | **4.39** | 9.75 | 86.06 | Ours | **85.81** | **83.81** | 91.83 | 91.12 | **4.28** | **2.43** |
| Change | +1.70 | +2.97 | +8.05 | +0.38 | +0.38 | +0.71 | Change (%) | +1.14 | +1.16 | -0.50 | -0.44 | +0.36 | +0.24 |

Table 6: Ablation experiments for *EEG* (left) and *ECG* (right) tasks

| Method | Sleep-EDF | | | | Method | Chapman | | | PhysioNet | | |
|---|---|---|---|---|---|---|---|---|---|---|---|
| | Acc ↑ | F1 ↑ | W-F1 ↑ | $\kappa$ ↑ | | Acc ↑ | F1 ↑ | AUC ↑ | Acc ↑ | F1 ↑ | AUC ↑ |
| $\mathcal{T}(\mathbf{x}, \phi)$ | 75.54±2.39 | 66.96±1.78 | 75.53±2.29 | 67.08±0.03 | $\mathcal{T}(\mathbf{x}, \phi)$ | 91.82 | 90.76 | 98.36 | 83.12 | 73.67 | 93.24 |
| $\mathcal{L}'_G$ | 77.21±1.51 | 67.67±1.67 | 76.89±1.71 | 69.39±0.02 | $\mathcal{L}'_G$ | 91.27 | 90.10 | 98.38 | 82.81 | 73.75 | 93.45 |
| $\hat{\mathcal{L}}_G$ | 77.75±1.23 | **68.04**±1.16 | **77.01**±1.07 | 69.94±0.01 | $\hat{\mathcal{L}}_G$ | 91.88 | 90.84 | 98.44 | **83.30** | 73.90 | **93.51** |
| Ours | **77.80**±1.95 | 67.01±2.65 | 76.77±2.58 | **70.01**±1.10 | Ours | **92.10** | **91.93** | **98.40** | 83.15 | **74.12** | 93.30 |
| Change | +2.26 | +0.05 | +1.24 | +2.93 | Change (%) | +0.28 | +1.17 | +0.04 | +0.03 | +0.45 | +0.06 |

i.e., without a guidance network for learning the mapping. The second row ($\mathcal{L}'_G$) represents the performance when the guidance network is only optimized using the categorical cross-entropy loss. The third row ($\hat{\mathcal{L}}_G$) presents the performance when the variance of angles is optimized to increase during training. And, the last row (Ours) is the original implementation of the proposed method. We also report the change when the mapping function is guided using the network $f_{G_\theta}$ and optimized using the loss defined in Equation 7, as opposed to being a fixed, non-learnable function.

As can be seen from the tables, when the models are trained by guiding the transformation function (with $f_{G_\theta}$), the performance of the models increases significantly up to 8%, except for the HHAR dataset with a marginal performance decrease of 0.5%. Importantly, adding the guidance network does not bring any additional parameters that help the learning, meaning that the model achieves improved generalization with the same capacity. Furthermore, the additional model parameters introduced to the overall framework approximately amount to one percent of those in the classifier.

While the performance increase can be associated with the decreased possible variations in the signals, our ablation experiments show that decreasing the variations blindly using the transformation with the same angle, decreases performance. Therefore, it is important to guide the transformation function for reducing the dimensionality, i.e., the space and time variations of a signal, of the whole data space. Overall, the results obtained from the ablation study and main experiments support the previous propositions and our motivation for introducing a novel diffeomorphism for preventing the inconsistency of deep learning models to the time shifts while increasing the generalization capability.

Table 7: Ablation experiments for *Audio*

| Method | Respiratory | | |
|---|---|---|---|
| | Acc ↑ | F1 ↑ | W-F1 ↑ |
| $\mathcal{T}(\mathbf{x}, \phi)$ | 21.28±7.43 | 55.03±2.89 | 18.14±6.39 |
| $\mathcal{L}'_G$ | 27.17±6.71 | 55.58±9.18 | 21.46±4.07 |
| $\hat{\mathcal{L}}_G$ | 28.57±8.31 | 54.28±6.56 | 23.73±4.65 |
| Ours | **33.10**±5.12 | **60.13**±4.67 | **28.33**±6.55 |
| Change | +11.82 | +5.10 | +10.19 |

Additional results (i.e., the extended experiments and ablations) regarding the performance of the proposed method can be found in Appendix D. Investigations regarding the performance improvements of the proposed diffeomorphism with different model networks are given in Appendix E. Detailed analysis of the guidance network with its effect is given in Appendix F. We provide an extended discussion of related work in Appendix G and outline limitations and future directions in Appendix H.

## 5 RELATED WORK

**Shift-invariant networks** Modern deep learning architectures use strided convolution or pooling to decrease the variance to a certain extent (Fukushima, 1980). However, Azulay and Weiss have demonstrated that a shift of one pixel in an image can lead to a significant alteration in the output probability of a trained classifier (Azulay & Weiss, 2018). Previous works showed that the downsampling caused aliasing and used low-pass filtering before the downsampling to prevent information loss (Zhang, 2019; Mairal et al., 2014). However, the used filters have suboptimal frequency responses, and realizing the ideal filter in practice is unfeasible. This leads to persistent aliasing, becoming a more significant concern for time series where high-frequency components are crucial for classification.

Adaptive subsampling methods have been recently explored for shift-invariance (Chaman & Dokmanic, 2021; Xu et al., 2021). Mainly, these methods perform subsampling on a constant (Chaman & Dokmanic, 2021) or input dependent (Rojas-Gomez et al., 2022) grid. This approach has a notable limitation in time series, particularly when nonlinear activation functions are involved. The methods tend to overlook variations in boundaries arising from the translation of samples, thereby imposing additional constraints on invariance (Rojas-Gomez et al., 2022). Consequently, the evaluation of

these methods is restricted to a narrow range of shifts, covering only a limited subset of the shift space. Moreover, their reliance on a grid scheme for sampling introduces a sensitivity to sampling rates, leading to performance gaps across the entire shift space (Michaeli et al., 2023). Therefore, we *shift the paradigm* and present a bijective transformation to modify the data space. Moreover, unlike existing methods that change network topology by modifying the pooling or adding extra filters without achieving complete shift-invariancy, our method guarantees invariance in neural network models without imposing any restrictions on the model topology or shift range.

**Time-delay neural networks**  Efforts to design shift-invariant models for time series predate modern deep learning methods (Hasegawa et al., 1996; Waibel et al., 1989). For example, a time-delay neural network (TDNN) network is designed to have the ability to represent relationships between events in time frames where the learned features by the network are aimed to be invariant under translation in time (Waibel et al., 1989). TDNN is trained with all time-shifted copies of samples and weights are updated by the average of all corresponding time-delayed error values. This is similar to the supervised training of a network with randomly shifted versions of samples. Although this strategy achieved shift-invariance for the first type of networks, as they do not include a pooling layer, it was shown that this approach is ineffective for modern architectures where pooling and derivatives are used (Scherer et al., 2010), and the network's invariance is limited to patterns seen during training and fails generalization (Azulay & Weiss, 2018). In this work, as we learn the mapping for each sample, the proposed transformation ensures that all shifted variants of a sample are mapped to a single point. Hence, a single data point effectively represents all the augmented variants.

## 6  CONCLUSION

The inadequacy of shift-invariance in deep learning models, particularly in the context of temporal data, remains a significant challenge. Existing solutions designed for images not only prove ineffective for time series but also result in performance deterioration for some tasks. To address this, we have introduced a novel differentiable bijective function. Our approach builds on the insight from Proposition 2.1, which states that the shift operation forms an Abelian group for each harmonic of a sample. Leveraging this property, we uniquely represent each point in the shift space using the phase angle of a harmonic whose period is at least as long as the sample length. Our approach ensures that samples, under various shifts, are mapped to a unique point in the data manifold without reducing dimensions, preserving task-related information without any loss. We validated our method theoretically and empirically, showing that it establishes shift-invariance in deep learning models without constraints on the shift range. In extensive experiments across six tasks, our approach consistently outperforms state-of-the-art methods, demonstrating its effectiveness in achieving complete shift-invariance without limitations on the model topology.

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

# APPENDIX

## A  THEORETICAL ANALYSIS

Here, we present complete proofs of our theoretical study, starting with notations. We assume all the samples (time series) are absolutely summable, and finite.

### A.1  REPRESENTATIONS AND NOTATIONS

#### A.1.1  FREQUENCY DOMAIN

Fourier transform of a real-valued sample with a finite duration is obtained as in Equation 10.

$$\mathcal{F}(\mathbf{x}) = |X(e^{j\omega})|e^{j\phi(\omega)} = \int_{-\infty}^{\infty} \mathbf{x}(t)e^{-j\omega t}, \tag{10}$$

where $\omega = \frac{2\pi}{T}$, and $\omega$ and T are the frequency in radian and period for all sinusoidals [3] in the range of Nyquist rate. $|X(e^{j\omega})|$ and $\phi(\omega)$ denote the amplitude and phase for all frequencies, respectively. Thus, the amplitude and phase angle of a particular sinusoidal are represented as $|X(e^{j\omega_0})|$ and $\phi(\omega_0)$. Similarly, the period for this sinusoidal is $T_0 = 2\pi/\omega_0$. The phase difference between a sinusoidal and an angle $\phi$ is shown in Equation 11.

$$\Delta\phi_{\mathbf{x}(t)} = \begin{cases} \frac{(\theta - 2\pi) * T_0}{2\pi}, & \text{if } \theta > \pi \\ \frac{\theta * T_0}{2\pi}, & \text{else} \end{cases}, \quad \text{and} \quad \theta = [\phi(\omega_0) - \phi] \% 2\pi, \tag{11}$$

where $T_0/2\pi = 1/\omega_0$. The calculated phase difference between the sample and given angle normalizes the phase change for the sinusoidal to exactly match the angle $\phi$ when shifted $\Delta\phi_{\mathbf{x}(t)}$ in the complex domain as in equations below.

$$\mathcal{F}(\mathbf{x}) = |X(e^{j\omega})|e^{j\phi(\omega)}e^{-j\omega\Delta\phi} \quad \text{after shifting} \quad |X(e^{j\omega})|e^{j\phi(\omega_0)}e^{-j\omega_0(\theta/\omega_0)} \tag{12}$$

$$|X(e^{j\omega_0})|e^{j\phi(\omega_0)}e^{-j\omega_0(\phi(\omega_0)-\phi)/\omega_0)} \tag{13}$$

$$|X(e^{j\omega_0})|e^{j\phi} \quad \text{for the sinusoidal with frequency } \omega_0 \tag{14}$$

#### A.1.2  TRANSFORMATION AND DIFFEOMORPHISMS

The observed samples $\mathbf{x}(t)$ from the sets with the (shift) variants are considered elements of manifolds $M^\phi$ according to the phase angle $\phi(\omega_0)$ of the harmonic with a period equal to or longer than the length of the segment $t$, i.e., the specific harmonic with period $T_0$ and frequency $\omega_0$. In other words, if the phase angle of the harmonic $\omega_0$ is $\phi_a$ for a sample $\mathbf{x}(t)$, the sample lies in manifold $\mathcal{M}^{\phi_a}$. The mapping function $f : \mathcal{M}^{\phi_a} \to \mathcal{M}^{\phi_b}$ between manifolds is defined as $\mathcal{T}(\mathbf{x}(t), \phi_b) = (\tilde{\mathbf{x}}(t), \Delta\phi_b)$ where $\mathbf{x}(t)$ lies in $\mathcal{M}^{\phi_a}$, and $\tilde{\mathbf{x}}(t)$ lies in $\mathcal{M}^{\phi_b}$. Similarly, the inverse mapping $f^{-1} : \mathcal{M}^{\phi_b} \to \mathcal{M}^{\phi_a}$ is represented as $\mathcal{T}(\tilde{\mathbf{x}}(t), \phi_a) = (\mathbf{x}(t), \Delta\phi_a)$.

$\tilde{\mathbf{x}}(t)$ and $\mathbf{x}(t)$ only differ by a random time shift $t'$ where this random time shift can be calculated using the phase angles of harmonics with frequency $\omega_0$, which makes the presented transformation a bijective function between time series manifolds. The differentiation of the mapping function at sample $\mathbf{x}(t)$ is shown as $Df_x : T_x\mathcal{M}^{\phi_a} \to T_{f_x}\mathcal{M}^{\phi_b}$.

---

[3]Harmonics and sinusoids are used interchangeably throughout the paper.

## A.2 NOTATION LIST

| Notation | Description |
|---|---|
| $\mathbf{x}$ | Time series represented as a bold lowercase symbol |
| $\mathcal{F}(\mathrm{x})$ | Discrete Fourier transformation of time series x |
| $\angle$ | The complex argument for obtaining phase values |
| $\omega$ | Variable that represents the frequencies in radian |
| $\lvert X(e^{j\omega}) \rvert$ | Magnitude components of the Fourier transformation of a time series |
| $\phi(\omega)$ | Phase components of the Fourier transformation of a time series |
| $\omega_0$ | Frequency of the specific harmonic whose period is equal or longer than the sample $\mathbf{x}$ |
| $\mathrm{T}_0 = \frac{2\pi}{\omega_0}$ | Period of the specific harmonic with frequency $\omega_0$ |
| $\Phi$ | Random variable for phase angles, i.e., $\phi \sim \Phi$ |
| $\phi(\omega_0)$ | Phase angle of a specific harmonic with the frequency $\omega_0$ |
| $\phi_{\mathbf{x}(t)}$ | Variable that represents phase angles of all harmonics for sample $\mathbf{x}(t)$ |
| $\phi_{\mathbf{x}(t-t')}$ | Variable that represents phase angles for sample $\mathbf{x}(t - t')$ |
| $\phi_{\mathbf{x}(t)}(\omega_0)$ | Phase angle of the specific harmonic with the frequency $\omega_0$ for sample $\mathbf{x}(t)$ |
| $\mathcal{T}(\mathbf{x}, \phi)$ | The proposed transformation function |
| $\Delta\phi$ | Phase difference between two angles |
| $\Delta\phi_{\mathbf{x}(t)}$ | Phase difference between the given angle $\phi$ in the transformation and the harmonic with frequency $\omega_0$ when sample $\mathbf{x}(t)$ is decomposed using Fourier transformation |
| $\Delta\phi_{\mathbf{x}(t-t')}$ | Phase difference between the given angle $\phi$ in the transformation and the harmonic with frequency $\omega_0$ when sample $\mathbf{x}(t - t')$ is decomposed using Fourier transformation |
| $\mathcal{M}^{\phi}$ | Manifold notation with an angle $\phi$ |
| $\mathcal{M}^{\phi_a}$ | A specific manifold with the angle $\phi_a$ |
| $\mathcal{T}(\mathbf{x}, \phi_a)$ | The output of the transformation, i.e., a time series that lies on the manifold $\mathcal{M}^{\phi_a}$ |
| $\mathcal{F}(\mathcal{T}(\mathbf{x}, \phi))$ | Fourier transformation of the output from the proposed transformation: Since the transformation function produces a tuple, we specifically apply the Fourier transformation to the first output, which corresponds to the time series. |
| $f_\theta(.)$ | Parametric mapping with parameter $\theta$ |
| $f_{G_\theta} : \mathbb{R}^d \to \Phi$ | The guidance network that outputs an angle to map the sample to a specific manifold |
| $f_{C_\theta} : X \to Y$ | The classifier neural network |
| $\mathrm{Var}(.)$ | Variance function |
| $\%$ | Modulo operation |

Table 8: Detailed list of notations used in this work

### A.3 PROOFS

**Lemma A.1** (Circular Shift). *Given a sample* $\mathbf{x}$ *in the interval* $[0, t_{int}]$ *and its shifted version* $\mathbf{x}(t-t')$, *where the shift is a random value from the finite real numbers, i.e.,* $t' \in (-\infty, \infty)$*. The shift is periodic with the signal length* $t_{int}$*, leading to the same vector representation when the shift is an integer multiple of the signal length.*

$$\infty > t > |t'| > -\infty, \ \ 0 = t' \ (\text{mod } t_{int}) \implies \mathbf{x}(t) = \mathbf{x}(t - t')$$

*Proof.* From circular shift, we know

$$\mathbf{x}(t) = \mathbf{x}(t \ (\text{mod } t_{int})) \tag{15}$$

$$\mathbf{x}(t - t') = \mathbf{x}((t - t') \ (\text{mod } t_{int})) \tag{16}$$

Therefore, $0 = t' \ (\text{mod } t_{int}) \implies \mathbf{x}(t) = \mathbf{x}(t - t')$ $\qquad\qquad\square$

Throughout the proofs, the length of the samples and their intervals are denoted by the same variable $t$. In other words, the samples are assumed to start at $t = 0$ and finish at $t_{int} = t$.

### A.4 PROOF FOR PROPOSITION 2.1

**Proposition A.2** (Time shift as a Group Operation). *Shift operation in time domain defines an Abelian Group of phase angles in the frequency domain for each harmonic with frequency* $\omega_k$.

$$(\Phi_k, + \ \text{mod } 2\pi), \ \ where \ \ \Phi_k = \phi \mid \phi = (\phi(\omega_k) + \omega_k t') \ mod \ 2\pi, t' \in \mathbb{R} \tag{17}$$

*Proof.* Let $\mathbf{x}(t + t')$ be randomly shifted variant of sample $\mathbf{x}(t)$ where $t' \in \mathbb{R}$. We can decompose these two sequences as in below using Fourier transformation.

$$\mathbf{x}(t) \xrightarrow{\mathcal{F}(.)} |X(e^{j\omega})|e^{j\phi(\omega)} \ \ \text{and} \ \ \mathbf{x}(t + t') \xrightarrow{\mathcal{F}(.)} |X(e^{j\omega})|e^{j(\phi(\omega)+\omega t')} \tag{18}$$

The phase values of these two time series for a harmonic at the given frequency can be expressed as follows.

$$\phi_{\mathbf{x}(t)}(\omega_k) = \phi(\omega_k) \ \ \text{and} \ \ \phi_{\mathbf{x}(t-t')}(\omega_k) = \phi(\omega_k) + \omega_k t' \tag{19}$$

Then, we can define a set of phase angles $\Phi_k$ with shift values $t'$.

$$\Phi_k = \{\phi \mid \phi = (\phi(\omega_k) + \omega_k t') \ \text{mod } 2\pi, t' \in \mathbb{R}\} \tag{20}$$

This set of phase angles with time shift operation defines the circle group $\mathbb{T}$. The circle group is Abelian (Fuchs, 1960), with time shifts corresponding to multiplication in the complex plane. For completeness, we have shown all the group axioms.

GROUP AXIOMS

The set of phase angles with time shift, $\Phi_k = \{\phi \mid \phi = (\phi(\omega_k) + \omega_k t') \ \text{mod } 2\pi, t' \in \mathbb{R}\}$, satisfies the five axioms of an Abelian group under modular addition.

*Axiom 1: Closure*
For any two phase angles $\phi_1, \phi_2 \in \Phi_k$, their sum is also in $\Phi_k$.

Let $\phi_1 = (\phi(\omega_k) + \omega_k t'_1) \ \text{mod } 2\pi$ and $\phi_2 = (\phi(\omega_k) + \omega_k t'_2) \ \text{mod } 2\pi$. Then their sum is:

$$\phi_1 + \phi_2 = (\phi(\omega_k) + \omega_k t'_1 + \phi(\omega_k) + \omega_k t'_2) \ \ \text{mod } 2\pi$$

$$\phi_1 + \phi_2 = (\phi(\omega_k) + \omega_k(t'_1 + t'_2)) \ \ \text{mod } 2\pi$$

Since $t'_1 + t'_2 \in \mathbb{R}$, the sum is also in $\Phi_k$, so closure holds.

*Axiom 2: Associativity*
For any three phase angles $\phi_1, \phi_2, \phi_3 \in \Phi_k$, their sum is associative under addition modulo $2\pi$.

$$((\phi_1 + \phi_2) \mod 2\pi) + \phi_3) \mod 2\pi = (\phi_1 + (\phi_2 + \phi_3) \mod 2\pi) \mod 2\pi$$

Let $\phi_1 = (\phi(\omega_k) + \omega_k t_1') \mod 2\pi$, $\phi_2 = (\phi(\omega_k) + \omega_k t_2') \mod 2\pi$, and $\phi_3 = (\phi(\omega_k) + \omega_k t_3') \mod 2\pi$. Then:

$$((\phi_1 + \phi_2) \mod 2\pi) + \phi_3 = (\phi(\omega_k) + \omega_k t_1' + \omega_k t_2') \mod 2\pi + \phi_3$$

$$= (\phi(\omega_k) + \omega_k(t_1' + t_2') \mod 2\pi + \omega_k t_3') \mod 2\pi$$

Similarly, for the right-hand side:

$$\phi_1 + ((\phi_2 + \phi_3) \mod 2\pi) = \phi_1 + (\phi(\omega_k) + \omega_k(t_2' + t_3') \mod 2\pi)$$
$$= (\phi(\omega_k) + \omega_k t_1' + \omega_k(t_2' + t_3')) \mod 2\pi$$

Using $(a+b) \mod 2\pi = ((a \mod 2\pi) + (b \mod 2\pi)) \mod 2\pi$, both sides simplify to $(\phi(\omega_k) + \omega_k(t_1' + t_2' + t_3')) \mod 2\pi$. Thus, associativity holds under addition modulo $2\pi$.

*Axiom 3: Identity Element*
The identity element in this group is the phase angle when no time shift has occurred, i.e., $t' = 0$.

$$\phi_0 = (\phi(\omega_k) + \omega_k \cdot 0) \mod 2\pi = \phi(\omega_k)$$

For any $\phi_1 \in \Phi_k$, we have:

$$\phi_1 + \phi_0 = \phi_1$$

Thus, $\phi_0$ is the identity element.

*Axiom 4: Inverse Element*
For any phase angle $\phi_1 = (\phi(\omega_k) + \omega_k t') \mod 2\pi$, its inverse is:

$$\phi_1^{-1} = (-\omega_k t') \mod 2\pi$$

Then:

$$\phi_1 + \phi_1^{-1} = (\phi(\omega_k) + \omega_k t' - \omega_k t') \mod 2\pi = \phi(\omega_k)$$

Thus, each element has an inverse.

*Axiom 5: Commutativity*
For any two phase angles $\phi_1, \phi_2 \in \Phi_k$, the sum is commutative:

$$\phi_1 + \phi_2 = \phi_2 + \phi_1$$

This follows from the commutativity of modular addition, so the group is Abelian. □

Proposition 2.1 states that shift operation in time domain defines an Abelian group of phase angles for each harmonic. Proposition 2.1 holds a key role in our algorithm, as it establishes an abstract connection between the time-domain shift operation and its effects on samples in the frequency domain.

### A.4.1 PROOF FOR THEOREM 2.2

**Theorem A.3** (Covering the Entire Time Space Injectively). *Given a sample* $\mathbf{x}$*, the defined function* $\mathcal{T}(\mathbf{x}, \phi) : \Phi \times \mathbb{R}^d \to \mathbb{R}^d \times \Delta\Phi$ *is bijective such that all shift variants of a sample can be covered with the unique phase angle of a harmonic whose period is longer or equal to the length of* $\mathbf{x}$*.*

$$\forall \phi_a, \phi_b \in \Phi, \ \mathcal{T}(\mathbf{x}, \phi_a) = \mathcal{T}(\mathbf{x}, \phi_b) \implies \phi_a = \phi_b$$
$$\forall t' \in \mathbb{R}, \ \exists \phi \in \Phi, \ \mathcal{T}(\mathbf{x}, \phi) = (\mathbf{x}(t - t'), \Delta\phi),$$

*Proof.*

$$\mathbf{x}(t) \xrightarrow{\mathcal{F}(.)} |X(e^{j\omega})| e^{j\phi(\omega)} \tag{21}$$

$$\mathbf{x}(t - t') \xrightarrow{\mathcal{F}(.)} |X(e^{j\omega})| e^{j(\phi(\omega) - \omega t')} \tag{22}$$

$$\phi_{\mathbf{x}(t)} = \phi(\omega), \ \ \phi_{\mathbf{x}(t-t')} = \phi(\omega) - \omega t' \tag{23}$$

$$\phi_{\mathbf{x}(t)} - \phi_{\mathbf{x}(t-t')} = \omega t', \tag{24}$$

Using Euler's formula,

$$\phi_{\mathbf{x}(t)} - \phi_{\mathbf{x}(t-t')} = (\omega t') \pmod{2\pi} \tag{25}$$

$$\phi_{\mathbf{x}(t)} - \phi_{\mathbf{x}(t-t')} = \left(\frac{2\pi}{T} t'\right) \pmod{2\pi} \tag{26}$$

$$\forall T_0 \in T, \ \infty > T_0 \geq t \implies \phi_{\mathbf{x}(t)}(\omega_0) - \phi_{\mathbf{x}(t-t')}(\omega_0) = \frac{2\pi}{T_0} t' \tag{27}$$

Thus,

$$\phi_{\mathbf{x}(t)}(\omega_0) - \phi_{\mathbf{x}(t-t')}(\omega_0) = \phi_{\mathbf{x}(t)}(\omega_0) - \phi_{\mathbf{x}(t-t')}(\omega_0) \implies t' = t' \tag{28}$$

Also,

$$\forall T_0 \in T, \ \infty > t > T_0 \implies \phi_{\mathbf{x}(t)}(\omega_0) - \phi_{\mathbf{x}(t-t')}(\omega_0) = \left(\frac{2\pi}{T_0} t'\right) \mod 2\pi, \tag{29}$$

$$\because \exists t' \in \mathbb{R}, \left(\frac{2\pi}{T_0} t'\right) > 2\pi \tag{30}$$

$$\therefore \ \exists t', t' \in \mathbb{R}, \ \neg(\phi_{\mathbf{x}(t)}(\omega_0) - \phi_{\mathbf{x}(t-t')}(\omega_0) = \phi_{\mathbf{x}(t)}(\omega_0) - \phi_{\mathbf{x}(t-t')}(\omega_0) \implies t' = t'), \tag{31}$$

which proves injection. Similarly, for surjection using Lemma A.1,

$$\forall t' \in \mathbb{R}, \ \infty > T_0 \geq t > |t'| \implies \phi_{\mathbf{x}(t)}(\omega_0) - \phi_{\mathbf{x}(t-t')}(\omega_0) = \frac{2\pi}{T_0} t' \tag{32}$$

$$\square$$

The final equation completes the proof by showing that the phase difference between the sample and its shifted version can get unique values for any shift, i.e., covering the whole time space.

A.4.2 PROOF FOR THEOREM 2.3

**Theorem A.4** (Guarantees for Shift-Invariancy). *Given a sample* $\mathbf{x}$ *and a randomly shifted variant of it* $\mathbf{x}(t - t')$, *if the transformation function* $\mathcal{T}(\mathbf{x}, \phi)$ *is applied to both samples with the same angle* $\phi_a$, *the resulting time series will be the same while carrying the same information.*

$$\mathcal{T}(\mathbf{x}(t), \phi_a) = \left(\tilde{\mathbf{x}}(t),\ \Delta\phi_{\mathbf{x}(t)}\right), \quad \mathcal{T}(\mathbf{x}(t - t'), \phi_a) = \left(\tilde{\mathbf{x}}(t),\ \Delta\phi_{\mathbf{x}(t-t')}\right)$$

*Proof.*

$$\mathbf{x}(t) \xrightarrow{\mathcal{F}(.)} |X(e^{j\omega})|e^{j\phi(\omega)}, \quad \mathbf{x}(t - t') \xrightarrow{\mathcal{F}(.)} |X(e^{j\omega})|e^{j\phi(\omega)}e^{\omega t'} \tag{33}$$

$$\phi_{\mathbf{x}(t)} = \phi(\omega), \quad \phi_{\mathbf{x}(t-t')} = \phi(\omega) - \omega t' \tag{34}$$

$$\phi_{\mathbf{x}(t-t')} - \phi_{\mathbf{x}(t)} = -\omega t' \tag{35}$$

Using Equation 11, the phase difference between samples $(\mathbf{x}(t), \mathbf{x}(t - t'))$ and $\phi_a$ can be obtained as,

$$\Delta\phi_{\mathbf{x}(t)} = \frac{[\phi(\omega_0) - \phi_a]}{2\pi} \cdot T_0, \quad \Delta\phi_{\mathbf{x}(t-t')} = \frac{[\phi(\omega_0) - \omega_0 t' - \phi_a]}{2\pi} \cdot T_0 \tag{36}$$

$$\Delta\phi_{\mathbf{x}(t-t')} - \Delta\phi_{\mathbf{x}(t)} = -\omega_0 t' \frac{T_0}{2\pi} \tag{37}$$

Using Fourier transform as in Equation 4,

$$\mathcal{F}\{\mathcal{T}(\mathbf{x}(t), \phi_a)\} = |X(e^{j\omega})|e^{j\phi(\omega)}e^{-j\omega\Delta\phi_{\mathbf{x}(t)}} \tag{38}$$

$$\mathcal{F}\{\mathcal{T}(\mathbf{x}(t - t'), \phi_a)\} = |X(e^{j\omega})|e^{j\phi(\omega)}e^{-j\omega t'}e^{-j\omega\Delta\phi_{\mathbf{x}(t-t')}} \tag{39}$$

Given that the amplitudes are identical, demonstrating equality in phase is sufficient, as shown below,

$$\phi_{\mathcal{T}(\mathbf{x}(t), \phi_a)} = \phi(\omega) - \omega\Delta\phi_{\mathbf{x}(t)}, \quad \phi_{\mathcal{T}(\mathbf{x}(t-t'), \phi_a)} = \phi(\omega) - \omega t' - \omega\Delta\phi_{\mathbf{x}_{(t-t')}} \tag{40}$$

$$\phi_{\mathcal{T}(\mathbf{x}(t-t'), \phi_a)} - \phi_{\mathcal{T}(\mathbf{x}(t), \phi_a)} = -\omega t' - \omega\Delta\phi_{\mathbf{x}_{(t-t')}} + \omega\Delta\phi_{\mathbf{x}(t)} \tag{41}$$

$$\phi_{\mathcal{T}(\mathbf{x}(t-t'), \phi_a)} - \phi_{\mathcal{T}(\mathbf{x}(t), \phi_a)} = -\omega t' - \omega\left[\Delta\phi_{\mathbf{x}_{(t-t')}} - \Delta\phi_{\mathbf{x}(t)}\right] \tag{42}$$

$$\phi_{\mathcal{T}(\mathbf{x}(t-t'), \phi_a)} - \phi_{\mathcal{T}(\mathbf{x}(t), \phi_a)} = -\omega t' - \omega\left[-\omega_0 t' \frac{T_0}{2\pi}\right] \tag{43}$$

$$\phi_{\mathcal{T}(\mathbf{x}(t-t'), \phi_a)} - \phi_{\mathcal{T}(\mathbf{x}(t), \phi_a)} = -\omega t' + \omega\left[\frac{2\pi}{T_0}t'\frac{T_0}{2\pi}\right] \tag{44}$$

$$\phi_{\mathcal{T}(\mathbf{x}(t-t'), \phi_a)} - \phi_{\mathcal{T}(\mathbf{x}(t), \phi_a)} = -\omega t' + \omega t' \tag{45}$$

$$\phi_{\mathcal{T}(\mathbf{x}(t-t'), \phi_a)} = \phi_{\mathcal{T}(\mathbf{x}(t), \phi_a)} \tag{46}$$

$$\mathcal{F}\{\mathcal{T}(\mathbf{x}(t), \phi_a)\} = |X(e^{j\omega})|e^{j\phi_{\mathcal{T}(\mathbf{x}(t), \phi_a)}} \tag{47}$$

$$\mathcal{F}\{\mathcal{T}(\mathbf{x}(t - t'), \phi_a)\} = |X(e^{j\omega})|e^{j\phi_{\mathcal{T}(\mathbf{x}(t-t'), \phi_a)}} \tag{48}$$

Therefore, the output time series samples will be the same after applying the transformation. $\qquad\square$

We complete the proof by showing the phase and magnitude of Fourier transformation of both samples are the same after transformation even though a random unknown shift is applied to the sample. Thus, the proposed transformation guarantees shift-invariancy without limiting the range of shifts.

# B    ALGORITHM

In this section, we present the pseudocode and the PyTorch (Paszke et al, 2019) implementation for the proposed transformation function. Algorithm 1 details each step of the transformation, which takes a sample, $\mathbf{x}$, and an angle, $\phi$ as inputs and outputs the transformed sample.

---

**Algorithm 1** Algorithm for the proposed diffeomorphism.

---

1: **Input:** $\mathbf{x}$, $\phi_a$

2: **Output:** $\mathcal{T}(\mathbf{x}, \phi_a)$

3: $|X(e^{j\omega})|e^{j\phi(\omega)} = \int_{-\infty}^{\infty} \mathbf{x}(t)e^{-j\omega t}$ ▷ Calculate the Fourier transformation to obtain harmonics

4: $\phi(\omega_0) = \angle\left(|X(e^{j\omega_0})|e^{j\phi(\omega_0)}\right)$ ▷ Obtain the angle for the harmonic with period $T_0$

5: $\theta = [\phi(\omega_0) - \phi_a] \% 2\pi$

6: $\Delta\phi = \begin{cases} \frac{(\theta - 2\pi) * T_0}{2\pi}, & \text{if } \theta > \pi \\ \frac{\theta * T_0}{2\pi}, & \text{else} \end{cases}$ ▷ Calculate the phase difference between the harmonic and the angle $\phi$

7: $|X(e^{j\omega})|e^{j(\phi(\omega) - \omega\Delta\phi)} = |X(e^{j\omega})|e^{j\phi(\omega)} * e^{-j\omega\Delta\phi}$ ▷ Apply a linear phase shift to each harmonic

8: **Return:** $\mathcal{F}^{-1}(|X(e^{j\omega})|e^{j(\phi(\omega) - \omega\Delta\phi)})$

---

Below, we provide the PyTorch implementation of our proposed transformation function, which includes two functions. The first function, `distanceCalculate`, computes the phase difference between the harmonic with frequency $\omega_0$ and the desired input angles. The second function, `diffeomorphism`, performs the main transformation: it takes as inputs the batch of samples $\mathbf{x}$ and the angles $\phi$, and outputs the transformed samples in the time domain.

```python
def distanceCalculate(angleDiff):
    theta = angleDiff % (2 * torch.pi)
    # Calculate the angular distance on the unit circle
    theta[theta > torch.pi] -= 2 * torch.pi
    return theta

def diffeomorphism(sample, desiredAngles):
    B, L, D = sample.shape
    samplesFFT = torch.fft.rfft(sample, dim=1)
    freq = torch.fft.rfftfreq(n=L)
    phAngle = torch.angle(samplesFFT)
    # Get the phase angle of the harmonic with frequency T_0
    angles = phAngle[torch.arange(phAngle.size(0)), 1, 0].squeeze()
    # Calculate the angle difference
    theta = distanceCalculate(angles-desiredAngles)
    # Normalize it to the sepecific harmonic with frequency w_0
    dtheta = theta / (2*torch.pi*freq[1])
    # Create complex exponentials with specific phase values
    linShift = torch.exp(-1j*2*torch.pi*freq[None,:]*dtheta[:,None])
    linShift = linShift.unsqueeze(dim=2).expand(-1, -1, D)
    # Apply a linear phase shift to all harmonics
    shiftedFFT = linShift*(samplesFFT)
    # Return to the time domain
    transformedSamples = torch.fft.irfft(shiftedFFT, n=L, dim=1)
    return transformedSamples
```

Implementation 1: PyTorch implementation of the proposed transformation

## C  EXPERIMENTS

Here, we give a detailed description of datasets, architectures, metrics, and training details for our experiments. We performed our experiments on NVIDIA GeForce RTX 4090 GPUs, involving training with three random seeds for all datasets, totaling approximately 480 GPU hours including ablation. We reported the mean of three runs with the standard deviation.

### C.1  DATASETS

In this section, we give details about the datasets that are used during our experiments. Overall, we have used eight datasets with six different time series tasks including *heart rate prediction*, *cardiovascular disease classification*, *activity recognition*, *step counting*, *sleep stage classification*, and *lung sound classification* from six sensor modalities *photoplethysmography*, *electrocardiogram*, *inertial measurement units*, *electroencephalography*, and *audio*. When selecting datasets for our experiments, we prioritized signals that provide meaningful insights into individuals' mental and physical health, where robust inference is particularly critical. Therefore, we specifically choose signals generated by humans. Additional to main experiments, in appendix, we also included *lung audio classification* from *audio* signals.

#### C.1.1  HEART RATE PREDICTION

**IEEE SPC**   The IEEE SPC dataset overall has 22 recordings of 22 subjects, ages ranging from 18 to 58 performing three different activities (Rocha et al., 2020). Each recording has sampled data from three accelerometer signals and two PPG signals along with the ECG data with a sampling frequency of 125 Hz. All these recordings were recorded from the wearable device placed on the wrist of each individual. All recordings were captured with two 2-channel PPGs with green LEDs, a tri-axial accelerometer, and a chest ECG for the ground-truth HR estimation. We averaged the two channels of PPG for prediction. We choose the last five subjects of SPC22 to be used for source domains. Throughout our experiments, we used PPG channels without integrating any inertial measurements.

**Dalia**   The PPG dataset for motion compensation and heart rate estimation in Daily Life Activities (DaLiA) was recorded from 15 subjects (8 females, 7 males, mean age of 30.6), where each recording was approximately two hours long. PPG signals were recorded while subjects went through different daily life activities, for instance sitting, walking, driving, cycling, working, and so on. PPG signals were recorded at a sampling rate of 64 Hz. The first five subjects are used as source domains, similar to Demirel & Holz (2023).

We standardize all PPG datasets as follows, same as the previous works (Biswas et al., 2019). Initially, a fourth-order Butterworth bandpass filter with a frequency range of 0.5–4 Hz is applied to PPG signals. Subsequently, a sliding window of 8 seconds with 2-second shifts is employed for segmentation, followed by z-score normalization of each segment. Lastly, the signal is resampled to a frequency of 25 Hz for each segment. We used an 8-block ResNet model with a stride of 2, a learning rate of $5e{-}4$, and a batch size of 32. Additional results with further experiments can be found in Appendix D.

#### C.1.2  HUMAN ACTIVITY RECOGNITION

**UCIHAR**   Human activity recognition using a smartphone's dataset (UCIHAR) (Anguita et al., 2012) is collected by 30 subjects within the age range of 16 to 48 performing six daily living activities with a waist-mounted smartphone. Six activities include walking, sitting, lying, standing, walking upstairs, and walking downstairs. Data is captured by 3-axial linear acceleration and 3-axial angular velocity at a constant rate of 50 Hz. We used the pre-processing technique the same as in (Qian et al., 2021) such that the input contains nine channels with 128 features (it is sampled in a sliding window of 2.56 seconds and 50% overlap, resulting in 128 features for each window). Windows are normalized to a mean of zero and unit standard deviation before feeding it to the models. The experiments are conducted with a leave-one-domain-out strategy with the first five subjects, where one of the domains is chosen to be the unseen target (Qian et al., 2022). We used an 8-block ResNet model with a stride of 2, a learning rate of $3e{-}3$, and a batch size of 32.

**HHAR** Heterogeneity Dataset for Human Activity Recognition (HHAR) is collected by nine subjects within an age range of 25 to 30 performing six daily living activities with eight different smartphones—Although HHAR includes data from smartwatches as well, we use data from smartphones—that were kept in a tight pouch and carried by the users around their waists (Stisen et al., 2015). Subjects then perform six activities including cycling, sitting, descending stairs, ascending stairs, standing, and walking. Considering the variable sampling frequencies of smart devices in the HHAR dataset, we downsampled the readings to 50 Hz. We employed sliding windows with lengths of 100 (two seconds) and 50, using a specified step size. These windows were then normalized to a mean of zero with unit standard deviation. In our experiments, we utilized the data from the first four subjects (i.e., a, b, c, d) as source domains, following a similar approach to previous papers (Qian et al., 2022; Demirel & Holz, 2023). We used an 8-block ResNet model with a stride of 2, a learning rate of $1e-3$, and a batch size of 64. The learning rate $1e-3$ for the guidance network, was the same for the activity recognition task.

### C.1.3 CARDIOVASCULAR DISEASE (CVD) CLASSIFICATION

**Chapman** Chapman University, Shaoxing People's Hospital (Chapman) ECG dataset which provides 12-lead ECG with a 10-second sampling rate of 500 Hz. The recordings are downsampled to 100 Hz, resulting in each ECG frame consisting of 1000 samples. The labeling setup follows the same approach as in Zheng et al. (2020) with four classes: atrial fibrillation, GSVT, sudden bradycardia, and sinus rhythm. The ECG frames are normalized to have a mean of 0 and scaled to have a standard deviation of 1. We split the dataset to 80–20% for training and testing as suggested in Zheng et al. (2020). We chose leads I, II, III, and V2 during our experiments for both ECG datasets.

**PhyioNet 2017** The 2017 PhysioNet/CinC Challenge aims to classify, from 8,528 single-lead ECG recordings (between 30 s and 60 s in length), whether the recording shows normal sinus rhythm, atrial fibrillation (AF), an alternative rhythm, or is too noisy to be classified, i.e., four classes. We normalize the signals to have zero mean and unit standard deviation. Additionally, we zero-pad the shorter recordings to ensure they have the same length. We split the dataset into training, validation, and test sets according to the patients using a 60, 20, 20 configuration.

For both datasets in CVD task, we used an 8-block ResNet model with a stride of 2, a learning rate of $5e-4$, and a batch size of 32. The learning rate for the guidance network was set same as the main classifier architecture.

### C.1.4 STEP COUNTING

The Clemson dataset has 30 participants (15 males, 15 females), Each participant wore three Shimmer3 sensors. We used the IMU sensor readings from non-dominant wrists to predict step count where each sensor recorded accelerometer and gyroscope data at 15 Hz. We calculated the total magnitude of the accelerometer and fed it to the model as a pre-processing without any filtering. We used window lengths of 32 seconds without an overlap in the regular walking setting. We conducted 10-fold cross-validation, with each fold consisting of 3 subjects for testing and validation. And, six randomly selected subjects were used for training in each fold. We used a 3 layer of FCN architecture, a learning rate of $5e-4$, and a batch size of 64. The learning rate for the guidance network was set same as the main classifier architecture.

### C.1.5 SLEEP STAGE CLASSIFICATION

We used the Sleep-EDF dataset which has five classes: wake (W), three different non-rapid eye movements (N1, N2, N3), and rapid eye movement (REM). The dataset includes whole-night PSG sleep recordings, where we used a single EEG channel (i.e., Fpz-Cz) with a sampling rate of 100 Hz. We employed the identical data split as presented in the paper (Eldele et al., 2021), accessible online, without applying any additional pre-processing steps. we used a 16-block ResNet model with a stride of 2, a learning rate of $1e-3$, and a batch size of 64. We ran three distinct seeds using the same split and reported the mean and standard deviation on the test set.

### C.1.6 Lung sound classification

We used Respiratory@TR which contains lung sounds recorded from left and right sides of posterior and anterior chest wall and back using two digital stethoscopes from 42 subjects collected in Antakya State Hospital (Altan et al., 2017). The 12 channels of lung sounds are focused on upper lung, middle lung, lower lung and costophrenic angle areas of posterior and anterior sides of the chest. The recordings are validated and labeled by two pulmonologists evaluating the collected chest X-ray, PFT and auscultation sounds of the subjects. Labels fall into 5 COPD severities (COPD0, COPD1, COPD2, COPD3, COPD4). The patients aged 38 to 68 are selected from different occupational groups, socio-economic status and genders. We performed 10-fold cross-validation on data from 42 subjects, with one fold reserved for validation in each iteration. All 12 channels, combining left and right chest wall recordings from six channels each, were used.

The audio was segmented into 8-second windows with a 2-second overlap to capture temporal patterns. We used an 8-block ResNet model with a stride of 3, a learning rate of $1e-5$, and a batch size of 15. Although audio models typically use Mel-frequency spectrograms with different networks, we did not observe a significant performance difference between these models and our initial model. Therefore, we used the original model with temporal data, consistent with our main experiments.

## C.2 Metrics

We used the common evaluation metrics in the literature for each task. Specifically, we used mean absolute error (MAE), root mean square error (RMSE), and Pearson correlation coefficient ($\rho$) for *heart rate prediction*. We used accuracy (Acc), macro-F1 score (F1) for activity recognition, and an additional area under the receiver operating characteristic curve (AUC) for cardiovascular disease classification (Kiyasseh et al., 2020). We used the mean absolute percentage error (MAPE) for step counting (Yang et al., 2020; Femiano et al., 2022). For lung audio classification, we evaluated performance using accuracy, macro F1, and weighted F1 scores (F1, W-F1). For sleep stage classification, we used the same metrics—accuracy, macro F1, and weighted F1 scores (F1, W-F1)—along with Cohen's Kappa coefficient ($\kappa$) (Cohen, 1960).

## C.3 Baselines

### C.3.1 LPF (Blurring)

In convolutional neural networks, pooling layers or convolutions with strides greater than 2 would conduct downsampling on the feature maps to reduce their size. However, since features are averaged or discarded in the downsampling, information may be lost, i.e., aliasing. Traditional pooling aggregates all values within the window to a single value. In contrast, Zhang (2019) aims to minimize information loss caused by pooling via replacing the traditional kernel with a Gaussian kernel as an low-pass filtering (LPF). Specifically, LPF applies a Gaussian-weighted function to the neighborhood values around each feature for convolution, and obtains a weighted average result.

Compared to the pooling operations of simply selecting the maximum/average value within the window, LPF uses Gaussian-weighted averages to preserve the relative positions and spatial relationships between features. In our implementations, we used 1D version of LPF with a length of 5. We evaluated filter lengths of $3, 5, 7$ to determine the optimum size for each task. During our experiments, we observed the best performance with a filter length of 5, except for HR prediction, where a length of 3 was optimal.

### C.3.2 APS

To make the sampling layer invariant to shifts, Chaman & Dokmanic (2021) proposed to subsample by partitioning feature maps into polyphase components and select the component with the highest norm. This approach has a significant limitation when applied to time series, especially with the nonlinear activation functions. It tends to overlook variations in boundaries arising from the translation of samples, thereby imposing additional shift constraints (Rojas-Gomez et al., 2022). Consequently, the evaluation of these methods is restricted to a narrow range of shifts, covering only a limited subset of the shift space. Moreover, as this approach selects the component with the highest norm, it requires feature maps to have unique values.

### C.3.3 WAVELET NETWORKS

Wavelet networks (Romero et al., 2024) consist of several stacked layers that respect scale and translation. At the beginning, the network consists of a lifting group convolution layer that lifts input time-series to the scale-translation group, followed by arbitrarily many group convolutional layers. At the end of the network, a global pooling layer is used to produce scale-translation invariant representations. Wavelet Networks are proposed for equivariant mappings between input and output. However, to turn an equivariant network into an invariant network, an extra layer that is equivariant in this degenerate sense (in practice, this often means either averaging or creating a histogram of the activations of the last layer) should be applied (Kondor & Trivedi, 2018). For example, the well-known wavelet scattering network achieves invariance by stacking equivariant layers followed by a final invariant one in that of scattering networks (Mallat, 2012). However, our proposed method does not require an additional layer as it operates on data manifolds directly.

We used the original GitHub (`https://github.com/dwromero/wavelet_networks`) implementation, leaving the dropout and base parameters unchanged. We searched over the learning rate (e.g., $\{10^{-3}, 10^{-4}, 10^{-5}\}$) using the validation set for each time series task.

### C.3.4 CANONICALIZATION

We also compared our method with a canonicalization approach which is based on learning mappings to canonical samples. Specifically, we used the equiadapt library (`https://github.com/arnab39/equiadapt`) (Mondal et al., 2023; Kaba et al., 2023) while adding translation equivarant architectures for shift operations. For translation equivariant canonicalization architecture, we implemented a convolutional neural network with a kernel size of five and three layers. The network avoids pooling layers to prevent aliasing and instead employs global average pooling at the final stage to aggregate information across the signal length. The output is reshaped to separate the channels corresponding to each discrete translation, followed by aggregation over the fiber channels to produce translation-equivariant activations.

We used a discrete Group representation with number of translations set to 16. The canonicalization network and classifier were trained jointly, incorporating a prior regularization loss to guide the learning process. For optimization, we employed the Adam optimizer with a learning rate of $1 \times 10^{-3}$ and no weight decay. We also performed a grid search to identify the optimal learning rate for the canonicalization network; however, no significant performance improvements were observed across different learning rates.

## C.4 IMPLEMENTATION DETAILS

Here, we have provided the details of the architectures, and hyperparameters. Primarily, we used the 1D ResNet (Hong et al., 2020) implementation in the supervised settings. While some alternative deep learning models can perform better in time series such as the combination of convolutional and LSTM layers, similar to previous works in shift consistency (Zhang, 2019), we focused on deep learning models which are mainly composed of convolutional layers.

### C.4.1 ARCHITECTURES

Here, we present the details of architectures that are investigated for the performance of shift-invariant techniques. Some details that are not given in the tables are as follows. Batch normalization (Ioffe & Szegedy, 2015) is applied after each convolutional block. ReLU activation is employed following batch normalization, in line with (He et al., 2015). We also applied a Dropout (Srivastava et al., 2014) with 0.5 after each activation and before the convolutions. Finally, a global average pooling is implemented before the linear layers.

Tables 9, 10a, and 10b give an overall for the architectures with the number of parameters for each dataset. From these tables, it can be observed that the number of parameters for the guidance network is much less than the main classifier, where the ratio is close to $\approx$2–4%.

The parameter count of the guidance network could be further reduced by selectively inputting only the important frequencies or the frequency band for each time series task, which can be determined using prior knowledge, rather than the entire spectrum. Nevertheless, for the sake of consistency, we

Table 9: ResNet architecture

| Repetition | Layer | Kernel Size | Output Size | Stride |
|---|---|---|---|---|
| 1 | Input (C,T) | - | (C, T) | - |
| 1 | Conv | (5, 1) | (64, T/2) | 1 |
| R | Residual Block | | | |
| | Conv | (5, 1) | (128, T/4) | S |
| | Conv | (5, 1) | (128, T/4) | 1 |
| 1 | Linear | - | (n_classes,) | - |

| # Parameters for *dataset* (C,T, R, S) | |
|---|---|
| *IEEE SPC* (C=1, T=200, R=8, S=2) | $\approx$210k |
| *DaLiA* (C=1, T=200, R=8, S=2) | $\approx$210k |
| *Chapman* (C=4, T=1000, R=8, S=2) | $\approx$197k |
| *PhysioNet* (C=1, T=6000, R=8, S=2) | $\approx$197k |
| *UCIHAR* (C=9, T=65, R=8, S=2) | $\approx$200k |
| *HHAR* (C=6, T=51, R=8, S=2) | $\approx$200k |
| *Respiratory* (C=12, T=6000, R=8, S=3) | $\approx$200k |
| *Sleep* (C=1, T=3000, R=16, S=2) | $\approx$3.2M |

Table 10: The model topologies of the classifier $f_C$ and guidance network $f_G$

(a) FCN architecture

| Layer | Kernel Size | Output Size |
|---|---|---|
| Input (C,T) | - | (C, T) |
| Conv (32 kernels) | (8, 1) | (32, T-4) |
| Max Pooling | (2,1) | (32, (T-4)/2) |
| Conv (64 kernels) | (8, 1) | (64, (T-4)/2-4) |
| Max Pooling | (2,1) | (64, (T-4)/4-2) |
| Conv (128 kernels) | (8, 1) | (128, (T-4)/4-6) |
| Max Pooling | (2,1) | (128, (T-4)/8-3) |
| Linear | - | (n_classes,) |

| # Parameters for *dataset* (C,T) | |
|---|---|
| *Clemson* (C=1, T=240) | $\approx$432k |

(b) Guidance architecture

| Layer | Kernel Size | Output Size |
|---|---|---|
| Input (C,T) | — | (C, T) |
| Conv (4 kernels) | (8, 1) | (4, T-5) |
| Max Pooling | (2,1) | (4, (T-5)/2+1) |
| Conv (16 kernels) | (5, 1) | (16, (T-5)/2-1) |
| Max Pooling | (2,1) | (16, (T-5)/4+1) |
| Conv (32 kernels) | (3, 1) | (32, (T-5)/4+1) |
| Max Pooling | (2,1) | (32, (T-5)/8+1) |
| Linear | - | (n_classes,) |

| # Parameters for *dataset* (C,T) | |
|---|---|
| *UCIHAR* (C=9, T=65) | $\approx$2.5k |
| *HHAR* (C=6, T=51) | $\approx$2k |
| *Clemson* (C=1, T=240) | $\approx$3k |
| *IEEE SPC* (C=1, T=200) | $\approx$2.4k |
| *DaLiA* (C=1, T=200) | $\approx$2.4k |
| *Chapman* (C=4, T=500) | $\approx$6k |
| *PhysioNet* (C=1, T=3000) | $\approx$13k |
| *Respiratory* (C=1, T=1500) | $\approx$66k |
| *Sleep* (C=12, T=6000) | $\approx$8k |

perform the Fourier transform with the number of harmonics same as the length of time series and provide the entire spectrum to the guidance network as the input.

One advantage of this input modeling is that the ratio between the number of parameters for the guidance network and the main classifier decreases more when the main classifier has more blocks as they are independent. For example, the guidance network has $1000\times$ fewer parameters than the main classifier for the sleep stage classification task where we have used 16 ResNet blocks for the classifier model for all techniques. At the same time, this addition increases the performance metrics up to 3%. Furthermore, increasing the number of parameters can decrease the performance of the models as it can cause overfitting of the training data (Cao et al., 2022; Wen et al., 2023), which is observed in our case as well (see Table 11 in Appendix D for additional results).

# D    ADDITONAL RESULTS

## D.1    SLEEP STAGE CLASSIFICATION

Here, we present extended results for the sleep stage classification in Table 11. Specifically, we include the F1 score as an additional metric and employ a larger network to observe its impact. Although our method ranked second in F1 metric, it is important to highlight that F1 scores are a

Table 11: Performance comparison of ours with other methods in *EEG* for sleep stage classification

| Method | Sleep-EDF | | | | |
| --- | --- | --- | --- | --- | --- |
| | S-Cons ↑ | Acc ↑ | F1 ↑ | W-F1 ↑ | $\kappa$ ↑ |
| Baseline | 95.06±0.61 | 75.41±2.01 | 65.40±1.33 | 74.87±1.92 | 67.12±2.96 |
| Baseline (2×) | 91.09±1.26 | 73.88±2.10 | 65.84±3.29 | 74.32±2.86 | 65.14±2.94 |
| Aug. | 99.00±0.17 | 74.89±1.11 | 64.71±1.55 | 74.03±1.46 | 65.89±1.81 |
| LPF | 92.43±1.24 | 73.56±2.93 | **68.08±1.97** | 76.01±1.98 | 65.68±3.46 |
| APS | — | — | — | — | — |
| Ours | **100±0.00** | **77.90±1.92** | 67.01±2.65 | **76.77±2.58** | **70.01±1.10** |
| Ours+LPF | 100±0.00 | 73.12±1.89 | 67.42±1.99 | 75.34±1.61 | 64.98±2.27 |

biased measure of classification quality (Christen et al., 2023; Powers, 2015), which is a problem when comparing recordings with a different prevalence of the classes as in sleep staging (Malafeev et al., 2018). Therefore, we reported additional metrics for measuring the performance. In particular, we included the kappa score, a metric widely used for evaluating algorithms in this task (Malafeev et al., 2018; Biswal et al., 2018). Additionally, sleep stage classification and certain other tasks have empty APS values as the authors' original implementation encountered overflow issues with matrix repetition, particularly for longer arrays.

Overall, our proposed method demonstrates a significant performance improvement in three of the metrics, with high kappa and accuracy—both of which are commonly used in the medical domain (Biswal et al., 2018) while ranking second in the F1 metric, following the LPF approach. We also performed the same ablation experiments for investigating the behavior of the loss function on the performance of the model for the sleep stage classification and reported the results in Table 12 while excluding the consistency metric as the model that include the proposed transformation are always completely shift-invariant.

Table 12: Ablation experiments for sleep stage classification

| Method | Sleep-EDF | | | |
| --- | --- | --- | --- | --- |
| | Acc ↑ | F1 ↑ | W-F1 ↑ | $\kappa$ ↑ |
| $\mathcal{T}(\mathbf{x}, \phi)$ | 75.54±2.39 | 66.96±1.78 | 75.53±2.29 | 67.08±0.03 |
| $\mathcal{L}'_G$ | 77.21±1.51 | 67.67±1.67 | 76.89±1.71 | 69.39±0.02 |
| $\hat{\mathcal{L}}_G$ | 77.75±1.23 | **68.04±1.16** | **77.01±1.07** | 69.94±0.01 |
| Ours | **77.80±1.95** | 67.01±2.65 | 76.77±2.58 | **70.01±2.50** |
| Change | +2.26 | +0.05 | +1.24 | +2.93 |

Table 12 also supports the previous experiments and claims regarding the advantages of guiding the proposed diffeomorphism with a neural network. When the models are trained by guiding the mapping function, the performance of the models increases up to 3% in Kappa ($\kappa$) score.

## D.2    HEART RATE PREDICTION

Here, we conducted additional experiments to evaluate the models' performance using varied amounts of training and testing data. Initially, we reduced the training data while increasing the testing data by dividing the datasets in half based on subjects to investigate the performance with less training data. Table 13 presents the results, where our proposed method also increases performance by 3–4% compared to the baseline architecture while reducing the variance between different runs and improving shift consistency by 40–60%, even in the low data regime.

Table 13: Performance comparison of ours and other methods in *PPG* datasets for HR estimation

| Method | IEEE SPC22 | | | | DaLiA | | | |
|---|---|---|---|---|---|---|---|---|
| | S-Cons (%) ↑ | RMSE ↓ | MAE ↓ | $\rho$ (%) ↑ | S-Cons (%) ↑ | RMSE ↓ | MAE ↓ | $\rho$ (%) ↑ |
| Baseline | 55.70±2.61 | 25.93±0.07 | 11.55±0.08 | 48.85±0.85 | 31.48±0.41 | 12.51±0.05 | 5.59±0.05 | 85.37±0.20 |
| Aug. | 70.35±0.47 | 26.21±0.48 | 12.06±0.33 | 48.50±1.22 | 53.38±0.29 | 12.31±0.09 | 5.60±0.05 | 85.90±0.13 |
| LPF | 67.43±0.56 | 25.87±1.02 | 14.03±0.69 | 47.76±2.49 | 39.82±1.33 | 12.65±0.09 | 5.81±0.01 | 84.87±0.23 |
| APS | 60.43±1.08 | 24.83±1.26 | 11.14±0.83 | 52.10±2.90 | 38.99±0.85 | 12.49±0.11 | 5.61±0.04 | 85.68±0.17 |
| Ours | 100±0.00 | **24.67±0.06** | **11.10±0.16** | **52.26±0.27** | 100±0.00 | **12.30±0.11** | **5.57±0.03** | **85.95±0.25** |
| Ours+LPF | 100±0.00 | 26.01±0.27 | 14.04±0.24 | 47.20±0.86 | 100±0.00 | 12.78±0.09 | 6.18±0.01 | 84.48±0.26 |
| Ours+APS | 100±0.00 | 24.67±0.32 | 11.38±0.13 | 51.64±0.73 | 100±0.00 | 12.40±0.08 | 5.67±0.02 | 85.63±0.30 |

We also performed the same ablation studies for heart rate prediction task in the low data regime and presented the results in Table 14.

Table 14: Ablation experiments for *HR* task with less training data

| Method | IEEE SPC22 | | | DaLiA$_{PPG}$ | | |
|---|---|---|---|---|---|---|
| | MAE ↓ | RMSE ↓ | $\rho$ ↑ | MAE ↓ | RMSE ↓ | $\rho$ ↑ |
| $\mathcal{T}(\mathbf{x}, \phi)$ | 12.04 | 25.49 | 50.72 | 6.10 | 12.94 | 85.03 |
| $\mathcal{L}'_G$ | 11.29 | 25.10 | 51.10 | 5.63 | 12.80 | 85.10 |
| $\hat{\mathcal{L}}_G$ | 11.32 | 25.08 | 51.18 | 5.60 | 12.71 | 85.12 |
| Ours | **11.10** | **24.67** | **52.26** | **5.57** | **12.30** | **85.95** |
| Change | +0.94 | +0.82 | +1.54 | +0.53 | +0.64 | +0.92 |

The ablation study results with a smaller training set align with our main findings, where reducing variations uniformly using the same angle negatively impacts performance. Likewise, expanding the potential solution space with $\hat{\mathcal{L}}_G$ leads to a performance decline compared to our method.

## D.3 DIFFEOMORPHISMS IN DEEP LEARNING

In this section, we review previous transformation functions and diffeomorphisms used in deep learning models. Invariant classification of input samples with neural networks has a long-standing history, with the Spatial Transformer Network (STN) being introduced to learn transformation functions for invariant image classification (Jaderberg et al., 2015). Similarly, Temporal Transformer Networks (TTN), an adaptation of STNs for time series applications, were introduced to predict the parameter of warp functions and align time series (Lohit et al., 2019; Shapira Weber & Freifeld, 2023). Recent methods have focused on optimizing a known family of diffeomorphism, known as diffeomorphic warping functions (Martinez et al., 2022) for time series alignment, through deep learning (Detlefsen et al., 2018; Shapira Weber et al., 2019). Thus, a significant distinction between our approach and previous techniques lies in the fact that we introduce a novel tailored diffeomorphism that is capable of mapping samples subjected to shifts to the same point in the high-dimensional data manifold, to ensure shift-invariancy.

Although previous techniques were designed for different purposes, such as time-warping (Lohit et al., 2019), we also evaluated and compared the performance and shift consistency of their transformation functions. Specifically, we implemented sequence temporal transformations (STN) for clinical time series from Oh et al. (2018) and TTN (Lohit et al., 2019) where the transformation and the classifier are trained together to maximize classification performance by minimizing the cross-entropy loss for both methods. In our implementation of STN, we followed the original design, using a neural network with two convolutional layers and two pooling layers, where pooling is applied between and after the convolutions. After the pooling operations, two fully connected layers are applied to the resulting feature maps to obtain the transformation parameter. The overall results in time series tasks are presented in the tables below.

Table 15: Performance comparison of our method with different transformations in HR estimation

| Method | IEEE SPC22 | | | | DaLiA | | | |
|---|---|---|---|---|---|---|---|---|
| | S-Cons (%) ↑ | RMSE ↓ | MAE ↓ | $\rho$ (%) ↑ | S-Cons (%) ↑ | RMSE ↓ | MAE ↓ | $\rho$ (%) ↑ |
| Baseline | 61.99±1.19 | 18.39±2.96 | 10.28±1.41 | 62.64±5.74 | 32.08±0.22 | 9.86±0.23 | 4.40±0.03 | 86.01±0.51 |
| Aug. | 76.48±1.77 | 18.73±1.15 | 10.42±0.40 | 64.06±3.70 | 52.77±0.39 | 9.85±0.21 | 4.47±0.06 | 85.99±0.49 |
| Baseline+STN | 67.13±1.53 | 18.45±2.73 | 10.35±0.73 | 63.49±4.10 | 44.81±0.25 | 9.90±0.17 | 4.43±0.04 | 85.96±0.47 |
| Baseline+TTN | 60.12±1.10 | 20.57±1.23 | 11.55±1.87 | 60.17±3.64 | 39.13±0.30 | 10.23±0.30 | 4.45±0.05 | 84.31±0.67 |
| Ours | **100±0.00** | **16.25±0.72** | **9.45±0.03** | **70.12±2.10** | **100±0.00** | **9.75±0.15** | **4.39±0.03** | **86.06±0.19** |

Table 16: Performance comparison of our method with different transformations in *ECG* datasets

| Method | Chapman | | | | PhysioNet 2017 | | | |
|---|---|---|---|---|---|---|---|---|
| | S-Cons (%) ↑ | Acc ↑ | F1 ↑ | AUC (%)↑ | S-Cons (%) ↑ | Acc ↑ | F1 ↑ | AUC ↑ |
| Baseline | 98.53±0.17 | 91.32±0.23 | 91.22±0.24 | 98.34±0.16 | 98.37±0.15 | **83.22±0.72** | 73.50±1.99 | 93.21±0.30 |
| Aug. | 99.00±0.16 | 91.96±0.19 | 91.89±0.22 | 98.45±0.18 | 98.96±0.17 | 82.28±1.18 | 72.32±2.20 | 93.20±0.42 |
| Baseline+STN | 98.31±0.13 | 91.45±0.20 | 91.33±0.19 | 98.31±0.14 | 98.55±0.14 | 83.12±0.50 | 73.27±1.54 | 93.23±0.28 |
| Baseline+TTN | 97.69±0.15 | 91.27±0.17 | 90.54±0.38 | 98.23±0.21 | 97.12±0.23 | 82.51±0.63 | 71.43±1.43 | 93.07±0.35 |
| Ours | **100±0.00** | **92.10±0.25** | **91.93±0.85** | **98.47±0.15** | **100±0.00** | 83.15±0.65 | **74.12±1.80** | **93.28±0.31** |

Table 17: Performance comparison of our method with different transformations in *IMU* datasets

| Method | UCIHAR | | | HHAR | | | Clemson | | |
|---|---|---|---|---|---|---|---|---|---|
| | S-Cons (%) ↑ | Acc ↑ | F1 ↑ | S-Cons (%) ↑ | Acc ↑ | F1 ↑ | S-Cons (%) ↑ | MAPE ↓ | MAE ↓ |
| Baseline | 94.07±1.38 | 85.39±2.30 | 83.20±2.94 | 98.27±0.33 | 91.87±1.36 | 91.16±1.38 | 54.31±4.40 | 4.76±0.11 | 2.74±0.08 |
| Aug. | 96.55±0.80 | 85.42±4.50 | 83.69±6.74 | 98.38±0.28 | **91.97±0.44** | **91.31±0.49** | 61.01±4.88 | 4.08±0.14 | 2.29±0.07 |
| Baseline+STN | 93.96±1.22 | 83.22±1.23 | 83.57±2.14 | 98.30±0.24 | 88.92±1.10 | 89.10±1.20 | 58.56±4.78 | 4.94±0.13 | 2.53±0.10 |
| Baseline+TTN | 93.32±1.95 | 83.27±1.57 | 82.78±3.12 | 97.10±0.78 | 90.03±1.74 | 90.18±1.10 | 45.89±3.02 | 5.43±0.20 | 2.89±0.18 |
| Ours | **100±0.00** | **87.71±1.98** | **85.67±2.47** | **100±0.00** | 91.93±1.14 | 91.12±1.03 | **100±0.00** | **4.28±0.34** | **2.43±0.21** |

As shown in the tables, other transformation functions fail to achieve true shift-invariance. While the STN method shows some improvements on certain datasets, it still lacks full shift-invariance. This limitation arises because STN relies on a neural network to estimate transformation parameters from time series data. However, this temporal transformation has no information about the position of the time series. Thus, when the input is shifted, the output changes.

## D.4 EXPANDED COMPARISON

Here, we compared the performance of techniques when random data augmentation (Aug.) is applied during training. In other words, the samples are randomly shifted and fed to the models during training. During inference, the performance of methods with original samples is evaluated without applying any shifts. We present the results in the tables below.

Table 18: Performance comparison of our method and other techniques with data augmentation for HR estimation

| Method | IEEE SPC22 | | | | DaLiA | | | |
|---|---|---|---|---|---|---|---|---|
| | S-Cons (%) ↑ | RMSE ↓ | MAE ↓ | $\rho$ (%) ↑ | S-Cons (%) ↑ | RMSE ↓ | MAE ↓ | $\rho$ (%) ↑ |
| Baseline | 61.99±1.19 | 18.39±2.96 | 10.28±1.41 | 62.64±5.74 | 32.08±0.22 | 9.86±0.23 | 4.40±0.03 | 86.01±0.51 |
| Aug. | 76.48±1.77 | 18.73±1.15 | 10.42±0.40 | 64.06±3.70 | 52.77±0.39 | 9.85±0.21 | 4.47±0.06 | 85.99±0.49 |
| LPF | 76.88±0.73 | 20.20±1.54 | 13.44±0.82 | 65.40±1.92 | 38.67±0.30 | 10.01±0.30 | 4.67±0.12 | 85.68±0.51 |
| APS | 73.99±1.06 | 19.42±0.60 | 12.98±0.29 | 65.27±1.32 | 44.33±0.16 | 10.45±0.40 | 5.01±0.17 | 84.69±0.85 |
| WaveletNet | 51.71±1.95 | 21.56±1.01 | 14.61±0.34 | 60.74±4.37 | 36.71±3.04 | 15.46±0.64 | 7.67±0.23 | 76.13±1.86 |
| LPF + Aug. | 76.17±1.15 | 20.39±1.15 | 13.22±0.36 | 61.72±2.87 | 55.62±0.21 | 12.57±0.19 | 6.02±0.11 | 84.94±0.40 |
| APS + Aug. | 74.88±0.61 | 18.01±0.15 | 10.57±0.15 | 66.40±2.21 | 53.70±0.08 | 12.84±0.10 | 6.08±0.02 | 85.62±0.65 |
| WaveletNet + Aug. | 50.14±0.14 | 20.10±1.15 | 13.41±0.57 | 61.90±3.50 | 36.71±3.04 | 15.46±0.64 | 7.67±0.23 | 76.13±1.86 |
| Ours | 100±0.00 | 16.25±0.72 | 9.45±0.03 | 70.12±2.10 | 100±0.00 | 9.75±0.15 | 4.39±0.04 | 86.06±0.19 |
| Ours+LPF | 100±0.00 | 20.34±1.62 | 13.77±0.84 | 65.60±2.31 | 100±0.00 | 10.72±0.11 | 5.30±0.03 | 84.12±0.23 |
| Ours+APS | 100±0.00 | 18.81±1.59 | 12.32±0.84 | 67.01±3.79 | 100±0.00 | 10.47±0.09 | 5.10±0.03 | 84.62±0.31 |

Table 19: Performance comparison of ours and other techniques with data augmentation in *ECG* datasets for CVD classification

| Method | Chapman | | | | PhysioNet 2017 | | | |
|---|---|---|---|---|---|---|---|---|
| | S-Cons (%) ↑ | Acc ↑ | F1 ↑ | AUC (%)↑ | S-Cons (%) ↑ | Acc ↑ | F1 ↑ | AUC ↑ |
| Baseline | 98.53±0.17 | 91.32±0.23 | 91.22±0.24 | 98.34±0.16 | 98.37±0.15 | 83.22±0.72 | 73.50±1.99 | 93.21±0.30 |
| Aug. | 99.00±0.16 | 91.96±0.19 | 91.89±0.22 | 98.45±0.18 | 98.96±0.17 | 82.28±1.18 | 72.32±2.20 | 93.20±0.42 |
| LPF | 98.69±0.14 | 92.01±0.23 | 91.94±0.58 | 98.50±0.24 | 98.94±0.39 | 84.40±0.16 | 75.68±0.76 | 93.80±0.32 |
| APS | 98.60±0.17 | 90.69±0.89 | 89.44±1.00 | 98.31±0.24 | — | — | — | — |
| WaveletNet | 91.02±1.14 | 90.87±1.02 | 90.02±1.00 | 97.94±0.21 | 65.03±0.71 | 76.06±0.64 | 63.35±3.40 | 87.02±0.29 |
| LPF + Aug. | 98.85±0.20 | 92.05±0.13 | 91.94±0.40 | 98.53±0.20 | 99.00±0.50 | 83.27±0.61 | 74.03±1.56 | 93.20±0.31 |
| APS + Aug. | 98.60±0.17 | 90.69±0.89 | 89.44±1.00 | 98.31±0.24 | — | — | — | — |
| WaveletNet + Aug. | 91.02±1.14 | 90.87±1.02 | 90.02±1.00 | 97.94±0.21 | 65.03±0.71 | 78.90±0.57 | 65.88±1.44 | 88.67±0.22/ |
| Ours | 100±0.00 | 92.10±0.25 | 91.93±0.85 | 98.47±0.15 | 100±0.00 | 83.15±0.65 | 74.12±1.80 | 93.28±0.31 |
| Ours+LPF | 100±0.00 | 92.05±0.52 | 91.96±0.54 | 98.51±0.10 | 100±0.00 | 85.20±0.40 | 77.50±1.21 | 94.20±0.19 |
| Ours+APS | 100±0.00 | 91.61±1.11 | 91.10±0.56 | 98.36±0.20 | — | — | — | — |

Table 20: Performance comparison of our method and others with data augmentation in *IMU* datasets for Activity and Step

| Method | UCIHAR | | | HHAR | | | Clemson | | |
|---|---|---|---|---|---|---|---|---|---|
| | S-Cons (%) ↑ | Acc ↑ | F1 ↑ | S-Cons (%) ↑ | Acc ↑ | F1 ↑ | S-Cons (%) ↑ | MAPE ↓ | MAE ↓ |
| Baseline | 94.07±1.38 | 85.39±2.30 | 83.20±2.94 | 98.27±0.33 | 91.87±1.36 | 91.16±1.38 | 54.31±4.40 | 4.76±0.11 | 2.74±0.08 |
| Aug. | 96.55±0.80 | 85.42±4.50 | 83.69±6.74 | 98.38±0.28 | 91.97±0.44 | 91.31±0.49 | 61.01±4.88 | 4.08±0.14 | 2.29±0.07 |
| LPF | 95.05±0.21 | 83.96±3.44 | 81.08±4.21 | 98.10±0.10 | 92.10±0.80 | 91.43±0.94 | 59.77±4.40 | 4.16±0.16 | 2.35±0.11 |
| APS | 96.40±0.03 | 81.75±4.11 | 79.01±5.33 | 98.30±0.24 | 91.83±1.35 | 91.01±1.47 | 45.50±2.69 | 4.74±0.16 | 2.69±0.07 |
| WaveletNet | 94.56±1.31 | 82.78±4.62 | 80.73±5.59 | 96.76±0.15 | 90.72±0.38 | 90.71±0.39 | 59.14±3.10 | 5.20±0.66 | 2.95±0.41 |
| LPF + Aug. | 97.65±1.30 | 84.67±3.45 | 83.32±3.50 | 98.65±0.12 | 92.45±0.78 | 91.70±0.62 | 59.77±4.40 | 3.81±0.13 | 2.23±0.10 |
| APS + Aug. | 96.40±0.03 | 78.40±3.75 | 75.43±4.33 | 98.87±0.34 | 92.40±0.40 | 91.49±0.73 | 45.50±2.69 | 3.94±0.10 | 2.50±0.07 |
| WaveletNet + Aug. | 94.56±1.31 | 82.78±4.62 | 80.73±5.59 | 96.76±0.15 | 90.72±0.38 | 91.71±0.39 | 60.23±2.54 | 5.18±1.02 | 3.02±0.48 |
| Ours | 100±0.00 | 87.71±1.98 | 85.67±2.47 | 100±0.00 | 91.93±1.14 | 91.12±1.03 | 100±0.00 | 4.28±0.34 | 2.43±0.21 |
| Ours+LPF | 100±0.00 | 84.78±2.46 | 82.58±2.62 | 100±0.00 | 92.51±0.55 | 91.80±0.62 | 100±0.00 | 3.75±0.33 | 2.12±0.18 |
| Ours+APS | 100±0.00 | 82.96±1.79 | 81.10±1.73 | 100±0.00 | 91.38±0.32 | 90.64±0.32 | 100±0.00 | 3.87±0.19 | 2.19±0.11 |

From the results, we can see that applying shift data augmentation during training did not consistently improve performance, and in some cases, it led to a decrease. For instance, in the HR prediction task, training the low-pass filtering method with randomly shifted samples resulted in lower performance. We believe that random shifts may reduce the inter-class separation between samples, causing them to overlap in the feature space (Wang et al., 2022). As a result, even though the number of training samples increases with augmentation, this reduced separation can lead to a decline in model performance.

## D.5 PERFORMANCE IN DIFFERENT HYPERPARAMETERS

In this section, we vary the training batch size to evaluate the performance of the proposed method. Given that the guidance network includes an additional loss function to reduce angle variance within a batch, we examined its performance under different batch sizes. The results are presented in Tables 21, 22, and 23.

Table 21: Performance comparison of our method and other techniques with a different batch size for HR estimation

| Method | IEEE SPC22 | | | | DaLiA | | | |
|---|---|---|---|---|---|---|---|---|
| | S-Cons (%) ↑ | RMSE ↓ | MAE ↓ | $\rho$ (%) ↑ | S-Cons (%) ↑ | RMSE ↓ | MAE ↓ | $\rho$ (%) ↑ |
| Baseline | 64.85±1.87 | 19.28±0.41 | 11.51±0.21 | 65.15±1.43 | 36.34±0.39 | 12.47±0.17 | 5.53±0.06 | 85.65±0.20 |
| Aug. | 78.90±1.12 | 19.51±0.93 | 11.82±0.49 | 63.62±1.55 | 52.77±0.39 | **12.36**±0.09 | 5.61±0.06 | 85.84±0.16 |
| LPF | 72.16±1.13 | 20.66±1.26 | 13.86±1.06 | 65.22±3.06 | 42.08±0.18 | 12.82±0.26 | 5.88±0.07 | 84.50±0.47 |
| APS | 71.25±1.19 | 19.79±0.92 | 12.59±0.68 | 67.05±0.85 | 42.01±0.26 | 12.37±0.18 | **5.50**±0.04 | **85.75**±0.39 |
| Ours | **100**±**0.00** | **18.29**±**0.71** | **11.30**±**0.57** | **69.10**±**1.20** | **100**±**0.00** | 12.49±0.16 | 5.55±0.07 | 85.60±0.23 |
| Ours+LPF | 100±0.00 | 20.49±0.77 | 14.18±0.42 | 67.28±3.20 | 100±0.00 | 13.17±0.23 | 6.43±0.11 | 83.63±0.57 |
| Ours+APS | 100±0.00 | 20.03±0.95 | 13.23±0.69 | 66.40±4.03 | 100±0.00 | 12.59±0.17 | 5.76±0.05 | 85.22±0.27 |

Table 22: Performance comparison of ours and other techniques with a different batch size in *ECG* datasets for CVD classification

| Method | Chapman | | | | PhysioNet 2017 | | | |
|---|---|---|---|---|---|---|---|---|
| | S-Cons (%) ↑ | Acc ↑ | F1 ↑ | AUC (%)↑ | S-Cons (%) ↑ | Acc ↑ | F1 ↑ | AUC ↑ |
| Baseline | 99.00±0.13 | 92.56±0.25 | **91.58**±0.26 | 98.59±0.12 | 98.41±0.26 | 82.71±1.55 | 73.16±3.39 | 93.00±0.30 |
| Aug. | 99.08±0.16 | 92.46±0.18 | 91.45±0.16 | 98.57±0.20 | 98.73±0.07 | 82.03±1.60 | 72.61±4.10 | 92.63±0.51 |
| LPF | 98.88±0.17 | 92.32±0.15 | 91.31±0.17 | 98.58±0.14 | 98.92±0.40 | 83.07±1.30 | 73.98±3.85 | 93.60±0.70 |
| Ours | **100**±**0.00** | **92.58**±**0.26** | 91.50±0.30 | **98.59**±**0.10** | **100**±**0.00** | **83.14**±**0.82** | **74.40**±**2.28** | **93.60**±**0.20** |
| Ours+LPF | 100±0.00 | 92.28±0.25 | 91.34±0.27 | 98.54±0.20 | 100±0.00 | 83.01±1.01 | 72.94±2.74 | 93.05±0.50 |

Table 23: Performance comparison of our method and others with a different batch size in *IMU* datasets for Activity and Step

| Method | UCIHAR | | | HHAR | | | Clemson | | |
|---|---|---|---|---|---|---|---|---|---|
| | S-Cons (%) ↑ | Acc ↑ | F1 ↑ | S-Cons (%) ↑ | Acc ↑ | F1 ↑ | S-Cons (%) ↑ | MAPE ↓ | MAE ↓ |
| Baseline | 95.13±1.21 | 89.89±1.76 | 89.37±1.82 | 98.25±0.14 | **91.99**±0.86 | **91.20**±0.91 | 43.61±1.53 | 4.27±0.26 | 2.41±0.14 |
| Aug. | 96.55±0.80 | 91.46±1.10 | 90.69±1.58 | 98.13±0.27 | 91.70±1.55 | 90.94±1.61 | 63.74±4.83 | **3.81**±0.21 | **2.13**±0.11 |
| LPF | 96.88±0.76 | 92.13±0.56 | 92.23±0.76 | 98.14±0.09 | 90.29±0.86 | 89.45±0.94 | 52.60±3.71 | 4.24±0.24 | 2.38±0.15 |
| APS | 97.12±0.75 | 92.43±1.40 | 92.07±1.21 | 98.30±0.24 | 91.83±1.35 | 91.01±1.47 | 42.16±0.98 | 5.00±0.03 | 2.84±0.02 |
| Ours | **100**±**0.00** | **92.78**±**0.51** | **92.94**±**0.33** | **100**±**0.00** | 91.77±0.56 | 91.10±0.60 | **100**±**0.00** | 4.06±0.16 | 2.24±0.10 |
| Ours+LPF | 100±0.00 | 89.78±1.25 | 90.04±1.11 | 100±0.00 | 91.40±1.20 | 90.60±1.32 | 100±0.00 | 4.14±0.09 | 2.32±0.06 |
| Ours+APS | 100±0.00 | 90.64±1.46 | 90.40±1.62 | 100±0.00 | 91.19±0.60 | 90.36±0.55 | 100±0.00 | 4.41±0.16 | 2.48±0.10 |

When we doubled the batch size during training, we observed a performance decrease in some datasets for our method, such as Clemson and DaLiA. We believe this could be due to the need to adjust the guidance network's learning rate when changing the batch size. Additionally, since the introduced loss function aims to reduce overall angle variance for a batch of samples, larger batches may pose optimization challenges for the guidance network. It is important to note that despite performance decreases in some cases, our method consistently outperformed baseline models with the original batch size used in the main experiments.

# E  IMPROVEMENTS IN PERFORMANCE ACROSS DIFFERENT NETWORKS

In this section, we conduct experiments to observe the performance of our proposed method when it is integrated into different network architectures. First, we employed the same 1D ResNet architecture for IMU related tasks as we used the fully convolutional network without residual connections in the main results because of better performance. Second, we applied a transformer with positional encoding, which is designed for time series tasks (Qian et al., 2022), to all tasks. Specifically, we used linear layers with a stack of four identical blocks. The linear layer converts the input data to embedding vectors of 128. Each block is made up of a multi-head self-attention layer and a fully connected feed-forward layer. We use residual connections around each layer.

We have not included blurring (LPF) and adaptive sampling in the transformer network analysis, as these methods are primarily tailored for convolutional architectures. Additionally, due to lack of convergence in the heart rate prediction task, we have omitted reporting results from the transformers. We reported the results in the tables below.

Table 24: Performance comparison of our method with others in *IMU* with ResNet

| Method | UCIHAR | | | HHAR | | | Clemson | | |
|---|---|---|---|---|---|---|---|---|---|
| | S-Cons (%) ↑ | Acc ↑ | F1 ↑ | S-Cons (%) ↑ | Acc ↑ | F1 ↑ | S-Cons (%) ↑ | MAPE ↓ | MAE ↓ |
| Baseline | 97.40±0.86 | 85.02±3.92 | 83.35±3.90 | 99.29±0.02 | 91.78±1.07 | 91.70±1.08 | 84.80±4.15 | 6.83±1.60 | 3.97±0.96 |
| Aug. | 98.40±0.37 | 86.66±1.26 | 85.12±1.53 | 99.31±0.04 | 92.82±0.35 | **92.84**±0.35 | 95.88±1.10 | 6.62±1.10 | 3.83±0.66 |
| LPF | 97.91±0.60 | 84.03±2.67 | 82.49±2.93 | 93.01±1.92 | 91.33±1.43 | 91.38±1.40 | 94.59±0.71 | 4.46±0.04 | 2.50±0.26 |
| APS | 98.02±0.46 | 81.98±3.36 | 79.23±4.20 | 93.01±1.92 | 92.13±0.22 | 92.14±0.22 | 93.01±1.92 | 6.61±1.44 | 3.84±0.84 |
| Ours | **100**±0.00 | **87.12**±2.21 | **85.21**±3.10 | **100**±0.00 | 91.90±0.10 | 92.02±0.07 | **100**±0.00 | 6.55±0.75 | 3.93±0.61 |
| Ours+LPF | 100±0.00 | 83.05±3.86 | 80.14±3.62 | 100±0.00 | **92.45**±0.45 | 92.50±0.44 | 100±0.00 | **4.45**±0.22 | **2.45**±0.13 |
| Ours+APS | 100±0.00 | 84.33±2.93 | 83.01±3.13 | 100±0.00 | 92.25±0.17 | 92.30±0.16 | 100±0.00 | 6.07±0.47 | 3.50±0.28 |

Table 25: Performance comparison of our method with others in *IMU* with Transformer

| Method | UCIHAR | | | HHAR | | | Clemson | | |
|---|---|---|---|---|---|---|---|---|---|
| | S-Cons (%) ↑ | Acc ↑ | F1 ↑ | S-Cons (%) ↑ | Acc ↑ | F1 ↑ | S-Cons (%) ↑ | MAPE ↓ | MAE ↓ |
| Baseline | 90.44±0.48 | 69.15±4.79 | 65.64±4.59 | 96.98±0.20 | 91.10±1.83 | 91.04±1.92 | 93.87±2.12 | 6.55±0.83 | 3.77±0.48 |
| Aug. | 93.68±0.64 | 73.23±2.75 | 69.79±3.63 | 98.35±0.06 | 89.21±0.07 | 89.16±0.11 | 95.46±2.65 | 6.54±0.34 | **3.76**±0.17 |
| Ours | **100**±0.00 | **74.02**±3.01 | **70.42**±3.47 | **100**±0.00 | **91.55**±1.20 | **91.19**±1.19 | **100**±0.00 | **6.50**±0.55 | 3.77±0.31 |

Table 26: Performance comparison of ours and others in *ECG* with Transformer

| Method | Chapman | | | | PhysioNet 2017 | | | |
|---|---|---|---|---|---|---|---|---|
| | S-Cons (%) ↑ | Acc ↑ | F1 ↑ | AUC (%) ↑ | S-Cons (%) ↑ | Acc ↑ | F1 ↑ | AUC ↑ |
| Baseline | 98.53±0.17 | 91.32±0.23 | 91.22±0.24 | 98.34±0.16 | 98.37±0.15 | 83.22±0.72 | 73.50±1.99 | 93.21±0.30 |
| Aug. | 99.00±0.16 | 91.96±0.19 | 91.89±0.22 | **98.45**±0.18 | 98.96±0.17 | 82.28±1.18 | 72.32±2.20 | 93.20±0.42 |
| Ours | **100**±0.00 | **92.10**±0.25 | **91.93**±0.85 | 98.40±0.15 | **100**±0.00 | **83.35**±0.65 | **74.12**±1.80 | **93.28**±0.31 |

We also integrated our proposed transformation into some recent neural networks and investigated the performance. Mainly, we employed ModernTCN (donghao & wang xue, 2024) and T-WaveNet (LIU et al., 2022) architectures. When we implemented ModernTCN, we follow the original implementation from We set the patch size and stride to 5 and 2, respectively, while keeping the backbone and dropout rate the same as in the original implementation. The stem, downsampling, and FFN ratios were set to 1.

Table 27: Performance comparison of our method for HR estimation using ModernTCN

| Method | IEEE SPC22 | | | | DaLiA | | | |
|---|---|---|---|---|---|---|---|---|
| | S-Cons (%) ↑ | RMSE ↓ | MAE ↓ | $\rho$ (%) ↑ | S-Cons (%) ↑ | RMSE ↓ | MAE ↓ | $\rho$ (%) ↑ |
| Baseline | 38.62±0.23 | 34.67±0.45 | 29.26±0.41 | 3.45±0.54 | 10.24±0.46 | 20.52±0.46 | 11.90±0.24 | 60.16±1.27 |
| Aug. | 50.15±0.07 | 34.36±2.14 | 28.29±3.04 | 01.95±2.23 | 39.10±1.27 | 15.36±1.16 | 7.25±0.64 | 79.97±2.53 |
| Ours | **100±0.00** | **33.45±1.07** | **29.33±1.09** | **08.96±3.73** | **100±0.00** | **15.20±1.22** | **7.13±0.10** | **80.05±1.08** |

Table 28: Performance comparison of our method for ECG datasets using ModernTCN

| Method | Chapman | | | | PhysioNet 2017 | | | |
|---|---|---|---|---|---|---|---|---|
| | S-Cons (%) ↑ | Acc ↑ | F1 ↑ | AUC (%) ↑ | S-Cons (%) ↑ | Acc ↑ | F1 ↑ | AUC ↑ |
| Baseline | 98.53±0.17 | 55.48±1.34 | 46.07±1.03 | 73.35±0.34 | 87.77±1.20 | 51.84±4.80 | 22.41±2.30 | 60.34±1.64 |
| Aug. | 95.51±3.38 | 81.93±3.60 | 79.15±4.63 | 94.88±1.11 | 98.96±0.17 | 60.13±2.57 | 28.86±5.23 | 70.57±6.10 |
| Ours | **100±0.00** | **83.80±2.06** | **80.73±2.70** | **95.41±0.33** | **100±0.00** | **60.58±1.50** | **30.73±1.08** | **72.58±3.76** |

Table 29: Performance comparison of our method for IMU datasets using ModernTCN

| Method | UCIHAR | | | HHAR | | | Clemson | | |
|---|---|---|---|---|---|---|---|---|---|
| | S-Cons (%) ↑ | Acc ↑ | F1 ↑ | S-Cons (%) ↑ | Acc ↑ | F1 ↑ | S-Cons (%) ↑ | MAPE ↓ | MAE ↓ |
| Baseline | 95.89±2.10 | 86.68±1.53 | 85.45±2.35 | 97.23±0.18 | 92.21±0.76 | 92.09±1.10 | 73.96±2.18 | 4.03±0.11 | 2.32±0.09 |
| Aug. | 96.55±0.80 | 85.42±4.50 | 83.69±6.74 | 98.38±0.28 | 91.97±0.44 | 91.31±0.49 | 61.01±4.88 | 4.08±0.14 | 2.29±0.07 |
| Ours | **100±0.00** | **88.73±1.47** | **87.19±1.94** | **100±0.00** | **94.12±0.87** | **93.43±1.10** | **100±0.00** | **3.88±0.26** | **2.15±0.16** |

Table 30: Performance comparison of our method using ModernTCN for sleep stage classification

| Method | Sleep-EDF | | | | |
|---|---|---|---|---|---|
| | S-Cons ↑ | Acc ↑ | F1 ↑ | W-F1 ↑ | $\kappa$ ↑ |
| Baseline | 71.84±1.78 | 69.50±1.81 | 62.84±1.39 | 71.21±1.66 | 60.39±2.20 |
| Aug. | 95.47±5.75 | 72.96±4.72 | 64.50±4.14 | 73.61±5.12 | 64.90±6.13 |
| Ours | **100±0.00** | **73.36±3.10** | **65.37±2.88** | **74.10±1.97** | **65.42±3.51** |

From these results, we can see that ModernTCN architecture performs relatively poor compared to 1D ResNet architecture in most tasks. However, for the IMU related tasks, ModernTCN outperforms other architectures. Similarly, when we integrate our method into the ModernTCN, the performance increases by 5–10% while decreasing the variation between runs.

We also performed experiments with T-WaveNet, a tree-structured wavelet deep neural network. The model decomposes input signals into multiple subbands and builds a tree structure with data-driven wavelet transforms the bases of which are learned using invertible neural networks. We use the original implementation from `https://openreview.net/forum?id=U4uFaLyg7PV`. Following the original implementation, the wavelet functions are learned together with the neural network instead of using stationary wavelet transforms like Haar.

Table 31: Performance comparison of our method for HR estimation using T-WaveNet

| Method | IEEE SPC22 | | | | DaLiA | | | |
|---|---|---|---|---|---|---|---|---|
| | S-Cons (%) ↑ | RMSE ↓ | MAE ↓ | $\rho$ (%) ↑ | S-Cons (%) ↑ | RMSE ↓ | MAE ↓ | $\rho$ (%) ↑ |
| Baseline | 49.73±1.61 | 21.78±1.87 | 15.77±1.59 | 60.30±4.16 | 39.69±0.17 | 13.38±0.19 | 6.15±0.08 | 83.10±0.47 |
| Aug. | 71.11±1.24 | 19.29±2.31 | 12.16±1.51 | 66.50±6.30 | 53.95±0.28 | 12.82±0.30 | 5.89±0.11 | 83.63±0.76 |
| Ours | **100±0.00** | **19.03±2.41** | **12.10±2.10** | **67.76±6.55** | **100±0.00** | 12.65±0.25 | **5.59±0.10** | **84.05±0.57** |

Table 32: Performance comparison of our method for ECG datasets using T-WaveNet

| Method | Chapman | | | | PhysioNet 2017 | | | |
|---|---|---|---|---|---|---|---|---|
| | S-Cons (%) ↑ | Acc ↑ | F1 ↑ | AUC (%) ↑ | S-Cons (%) ↑ | Acc ↑ | F1 ↑ | AUC ↑ |
| Baseline | 97.23±0.19 | 93.17±0.80 | 92.40±0.78 | 98.87±0.18 | 94.72±1.91 | 78.94±1.60 | 71.43±2.24 | 92.44±1.03 |
| Aug. | 98.19±0.68 | **93.63±0.36** | 92.89±0.32 | 98.96±0.10 | 95.43±0.89 | 79.77±0.99 | 70.72±1.42 | 92.33±0.63 |
| Ours | **100±0.00** | 93.45±0.40 | **92.92±0.50** | **99.00±0.15** | **100±0.00** | **79.89±1.10** | **71.61±2.10** | **92.50±0.89** |

Table 33: Performance comparison of our method for IMU datasets using T-WaveNet

| Method | UCIHAR | | | HHAR | | | Clemson | | |
|---|---|---|---|---|---|---|---|---|---|
| | S-Cons (%) ↑ | Acc ↑ | F1 ↑ | S-Cons (%) ↑ | Acc ↑ | F1 ↑ | S-Cons (%) ↑ | MAPE ↓ | MAE ↓ |
| Baseline | 97.63±1.45 | 72.77±2.36 | 70.48±4.16 | 98.37±0.96 | 92.37±0.89 | 91.53±1.03 | 89.50±0.50 | 6.69±0.54 | 3.90±0.31 |
| Aug. | 98.30±2.43 | 72.82±3.34 | 68.78±3.53 | 98.68±0.65 | 92.88±1.15 | **92.05±1.28** | 89.29±0.72 | 6.62±0.59 | 3.83±0.31 |
| Ours | **100±0.00** | **74.04±2.10** | **71.04±3.25** | **100±0.00** | **92.95±1.14** | 91.60±0.93 | **100±0.00** | **6.03±0.50** | **3.54±0.35** |

Table 34: Performance comparison of our method using T-WaveNet for sleep stage classification

| Method | Sleep-EDF | | | | |
|---|---|---|---|---|---|
| | S-Cons ↑ | Acc ↑ | F1 ↑ | W-F1 ↑ | $\kappa$ ↑ |
| Baseline | 71.84±1.78 | 69.50±1.81 | 62.84±1.39 | 71.21±1.66 | 60.39±2.20 |
| Aug. | 95.47±5.75 | 72.96±4.72 | 64.10±1.23 | 73.61±5.12 | 64.90±6.13 |
| Ours | **100±0.00** | **73.36±5.10** | **65.90±1.07** | **74.10±3.97** | **65.42±3.51** |

As shown in tables, the proposed transformation also increases the performance of the different neural networks. One important result from the comparison of these tables is that there is a correlation between the model's ability to remain invariant to shifts and its performance, up to a certain threshold where the model performs adequately. However, beyond that point, as the model's performance declines, the consistency in shift increases, resulting in the model consistently outputting the wrong class. For instance, in the case of step counting, the transformer architecture performs quite worse and fails to distinguish between samples. As a result, the consistency of the transformer is higher compared to ResNet and fully convolutional networks while the performance is lower. We believe that investigating the invariance of different neural architectures alongside their performance on time series tasks can shed light on model networks and the fundamental reasons behind abrupt output changes with small changes in the input signal, i.e., $\approx$ 10–15 ms shift.

## F    VISUAL EXAMPLES FOR THE GUIDANCE NETWORK

In this section, we provide some visual examples to show how the proposed transformation function works. First, we show the t-SNE (van der Maaten & Hinton, 2008) representations of the embeddings obtained from a trained model with and without applying our transformation in Figure 4.

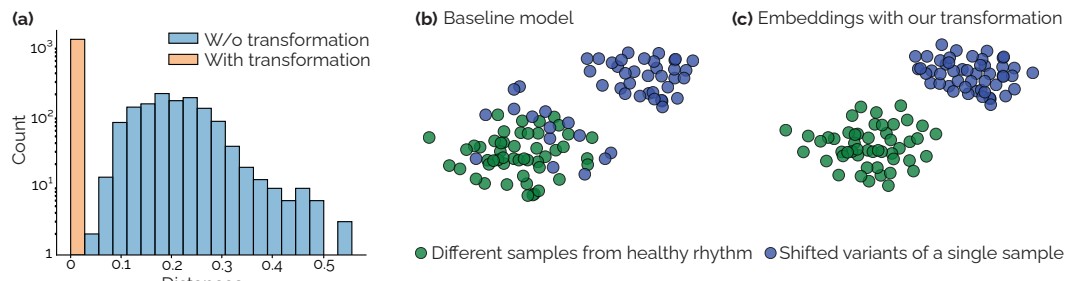

Figure 4: **(a)** Comparison of pairwise Euclidian distances of randomly shifted embeddings with and without applying our method. **(b)** t-SNE visualizations of embeddings without our method show some shifted samples clustering with opposite class embeddings. **(c)** With our transformation, all shifted variants of the same signal cluster correctly within their true class label.

For visualization, we selected 50 different ECG (healthy) signals from the test set. We then took a single arrhythmia ECG sample from the test set, applied 49 shifts to it (50 samples with the original), and created variants shown in blue. Finally, we compared the embeddings with and without applying our proposed transformation function. As seen in Figure 4, applying our transformation function maps the shifted samples to a single point in the embedding space, with the maximum Euclidean distance between embeddings being close to $10^{-6}$.

Second, we conducted a simple experiment to investigate how the guidance network works with the proposed transformation. Specifically, we created a two-label classification task where the model classifies sinusoids by frequency. The dataset includes two waveforms: $x_1(t) = \cos(\omega_1 t + \phi_1) + \cos(\omega_2 t + \phi_2)$ and $x_2(t) = \cos(\omega_1 t + \phi_1) + \cos(\omega_3 t + \phi_3)$, with the model identifying whether the input contains frequency $\omega_2$ or $\omega_3$. Frequencies were set at 5, 24, and 25 Hz for $\omega_1$, $\omega_2$, and $\omega_3$, respectively, with $\omega_1$ included in both waveforms to increase task difficulty. We set the sampling rate of the signals to 300 Hz.

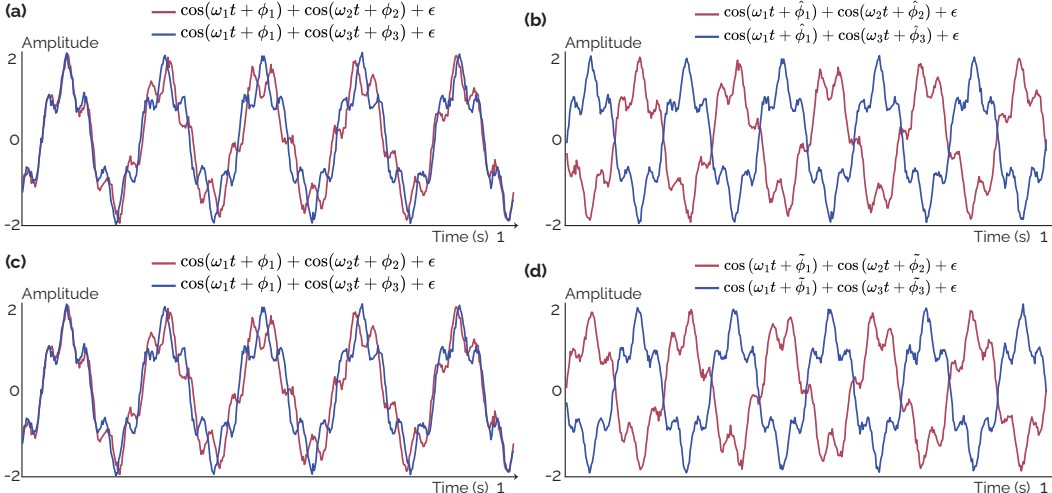

Figure 5: Input waveforms to the classifier in the third epoch **(a)** without applying our transformation function. **(b)** with our guidance network ($f_{\theta_G}$). Another experiment with a different seed. Input waveforms **(c)** without applying our transformation function. **(d)** with our guidance network ($f_{\theta_G}$).

To add diversity, we randomly shifted $x_1(t)$ and $x_2(t)$ by angles sampled from $[0, \pi)$ and added Gaussian noise ($\epsilon$) with variance of 0.1. We used the FCN similar to that specified in Appendix C.4 as the architecture. This experimental setup is inspired by similar experiments exploring neural network behaviors (Rahaman et al., 2019).

Figure 5 illustrates an interesting result: after a few weight updates, the guidance network assigns angles $\phi$ that maximize the Euclidean distance between inter-class samples. For instance, before applying the guidance network, the distance between a pair of samples is 9.12 (Figure 5 (a)). After applying the guidance network, this distance increases by four to 43.6 (Figure 5 (b)). Interestingly, running the same experiment with a different seed (i.e., a new random initialization of the guidance network) shows that the assigned angles differ, but the Euclidean distances between the samples remain almost unchanged, shifting only slightly from 43.6 to 42.8 (See Figure 5 (d)).

This experiment can also explain the occasional performance increase when the angle variance increases with a loss term as there is no single solution for minimizing the distance, but there can be infinitely many depending on the frequency distribution of the dataset and classes.

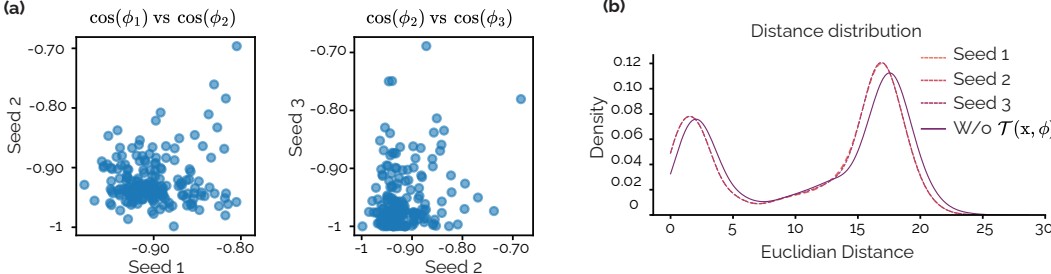

Figure 6: **(a)** Angle assignments across different seeds. **(b)** Euclidean distances between intra-class samples, compared with and without the proposed transformation (W/o $\mathcal{T}(\mathbf{x}, \phi)$). The results show that the Euclidean distances are highly consistent across different seeds, with the curves nearly overlapping. Additionally, when the transformation is not applied, the distances between intra-class samples are noticeably higher.

We conducted an additional experiment to analyze the angle assignments across runs using the IEEE SPC22 dataset. The experiment was performed with three random seeds, and the results are presented in Figure 6. In Figure 6 **(a)**, we reported the assigned angles in $\cos(\phi)$ as the angle $-\pi$ and $\pi$ are the same due to the circular property of the angles. While the assigned angles vary slightly between runs, the Euclidean distances between samples consistently converge to similar values. Specifically, the intra-class samples are closer in the transformed space compared to the case when the proposed transformation is not applied.

## G EXPANDED REVIEW OF RELATED WORK

**Transformation**   Applying phase shifts to time series is commonly used in signal processing (Haykin & Veen, 2002; Oppenheim et al., 1996). Recently, shifting phase values of harmonics have also been applied in the machine learning community for data augmentation of time series (Demirel & Holz, 2023; Qian et al., 2022). However, our proposed transformation differs from these in two key aspects. First, our work is the first to represent every point in the shift space uniquely with the phase angle of a harmonic whose period is equal to or longer than the length of the sample, i.e., $T_0 \leq t$. This observation enables us to design a bijective transformation that ensures shift invariance. Additionally, we integrated this observation into deep learning frameworks using a novel loss function, demonstrating that our proposed method enhances model performance while ensuring shift invariance.

Second, we apply linear phase shifts to keep waveform features intact. Since if an input signal is subjected to a phase shift that is a nonlinear function of $\omega$ similar to Qian et al. (2022); Demirel & Holz (2023), then the complex exponential components of the input at different frequencies will be shifted in a manner that results in a change in their relative phases. Superimposing these exponentials can result in a signal that significantly differs from the input if special precautions are not taken. Thus, this alteration in the waveform (Oppenheim et al., 1996) can potentially affect downstream labels or generate unrealistic signals.

However, our transformation applies the tailored shift linearly to all harmonics while keeping the information content unchanged (Mallat, 2012) as the transformation operates as a group action.

**Shift-invariant Kernels**   Learning shift invariant representations from data has a long history in machine learning (Grosse et al., 2007; Rahimi & Recht, 2007). The initial effort focused on designing shift-invariant kernels for feature extraction (Rahimi & Recht, 2007), which were applied to support vector machines. A different approach introduced shift-invariant sparse coding technique, which reconstructs an input using all basis functions across all possible shifts (Grosse et al., 2007). However, the classification performance of these techniques were significantly outperformed by the modern networks. Thus, recent approaches have focused on integrating shift-invariant kernels into modern convolutional neural networks in a stacked manner while using Gaussian low-pass filters to prevent aliasing (Mairal et al., 2014).

However, applying low-pass filters to prevent anti-aliasing completely is not possible (Oppenheim et al., 1996). Thus, high-frequency components will always (partially) alias. This is more problematic for time series as the interaction of high and low frequencies are more common (Demirel & Holz, 2023; Canolty & Knight, 2010). Therefore, applying a low-pass filter can reduce the performance of neural networks in certain time series tasks, as shown by our experiments. The filtering may inadvertently attenuate important high-frequency components, which are essential for distinguishing patterns, leading to suboptimal model outcomes.

**Learning based Transformations**   Methods to standardize inputs have been around for a long time (Yüceer & Oflazer, 1993). An important recent work along this direction is the Spatial Transformer Network (STN) being introduced to learn transformation functions for invariant image classification (Jaderberg et al., 2015). Similarly, Temporal Transformer Networks (TTN), an adaptation of STNs for time series applications, were introduced to predict the parameter of warp functions and align time series (Lohit et al., 2019; Shapira Weber & Freifeld, 2023). Recent studies have utilized canonical equivarant networks to obtain mapping points for inputs (Kaba et al., 2023). However, these methods face significant limitations as the operation order increases. Specifically, higher-order transformations in group equivarant networks require additional filter copies in the lifting layer and an increased number of parameters in the subsequent group convolution layers. While this can improve performance, it comes at the cost of significantly higher computational and model complexity. Furthermore, prior works restrict mappings to a finite number of group elements defined by the canonicalization network. In contrast, our proposed transformation eliminates this limitation entirely, enabling each sample to map to any point in the input space—infinitely many—without relying on a neural network. This is achieved by uniquely representing each point in the shift space using a specific harmonic.

## H    DISCUSSION, LIMITATIONS AND FUTURE WORK

In this work, we propose a new diffeomorphism to achieve shift invariant deep learning models for time series in real-world tasks. While existing techniques show promise for images, they fall short in time series, where the interaction of low and high frequencies are an important part of the data generation. The proposed transformation offers a novel solution, ensuring that samples will map the same point in the high dimensional data manifold despite a random shift. Theoretical and empirical analysis demonstrates its effectiveness across several time series tasks, enhancing model robustness while improving the performance.

While our approach consistently improves the performance of deep learning models for time series data, it is worth noting the potential areas for future investigation and improvement. First, we conducted our experiments on health-related time series tasks from humans since the robustness of models is crucial in those domains. Therefore, extending the proposed transformation to images for shift or rotation invariancy presents an intriguing direction for future investigations. Thus, we believe that future research could benefit on adapting our approach to diverse domains, including images, to explore shift or rotation invariance further. Second, our approach requires samples to be expanded into the sum of periodic sinusoidals with Fourier expansion followed by using the phase angle of the one whose period equals or exceeds the length of the signal. Input samples should be decomposed the sinusoidals while considering this requirement. Therefore, we believe future work can benefit by detecting the phase of a specific sinusoidal which satisfies the condition and apply a linear phase all-pass filter without performing the operation in the frequency domain. Lastly, we observed notable performance improvements from the additional guidance network when it is used with the proposed diffeomorphism while applying a proper loss function. Thus, we believe that further performance improvements can be achieved through a refined design incorporating alternative inputs.

