# OpenReview forum: "Shifting the Paradigm: A Diffeomorphism Between Time Series Data Manifolds for Achieving Shift-Invariancy in Deep Learning"
_ICLR.cc/2025/Conference — ICLR 2025 Poster_

### Official Review · Reviewer_Se81 · 2024-10-31

**Soundness:** 2
**Presentation:** 3
**Contribution:** 2
**Rating:** 6
**Confidence:** 3

**Summary:**

The paper introduces a novel method to achieve shift-invariance in deep learning models for time series data. Traditional methods often lack robustness to shifts, which can lead to highly inconsistent performance in tasks such as heart rate prediction, especially when requiring both low- and high-frequency information. The authors propose a differentiable bijective transformation that maps shifted samples to a unique point in a high-dimensional manifold, ensuring shift-invariance without dimensional reduction or architectural changes. This approach is tested across various time series tasks, where it demonstrates improved performance and consistency compared to existing techniques.

**Strengths:**

The paper is well-written, and the figures all look aesthetically nice and neat.
The medical/empirical motivation is clearly laid out, with good intuitive examples of why this is an important and challenging problem.
The empirical results appropriately cover a range of real-world empirical datasets with a range of important tasks.

**Weaknesses:**

- The authors present the proposed transformation as a novel aspect of the method, yet phase-shifting is a well-known approach in signal processing. The architectural novelty here lies primarily in the integration of the guidance network, not the phase-shifting transformation itself. The paper however does not provide an independent evaluation of the guidance network’s specific performance in estimating the phase angle \phi, which could easily be done when ground truth shifts are known. Most evaluations, including the ablation studies, focus on downstream tasks, which are interesting and practically relevant but do not provide clear insight into what the algorithm is doing at the shift-invariance level. They also add confounding factors. For instance, in the appendix, you argue that "it can be observed that the number of parameters for the guidance network is much less than the main classifier", around 2% of the number of parameters, but the classifier is not an inherent part of your shift-invariant approach.

- A natural follow-up question is how well the guidance network performs in OOD settings where the ground truth shifts are known but the time series were not part of the training data. Since the learned latent representations are still just time series, a simple illustration of the approach could be to take in a number of shifted time series and visualize the resulting latent representation of the time series.

- While the shift consistency captures something in this direction, it still requires training the classifier to assign class probabilities. While the S-Cons results look impressive across the experimental results, could one not directly measure the variance or consistency of the output angles (or phase-shifted latent representations) across the shifted samples to get clearer insights into what is happening here?

- In that spirit, there could also be comparisons to other approaches in time series alignment. This could also provide a more direct evaluation of the shift-invariant transformation and isolate the main contribution of the method.

In summary, while the paper introduces an interesting approach, these concerns prevent me from fully supporting acceptance in the current form. But I am open to adjusting my assessment if some potential confusion is clarified/some clarifying results are provided.

**Questions:**

- Figure 1: while I appreciate the thorough overview, the figure contains a lot of information that is a bit hard to grasp without more detailed descriptions or additional reading, and connections between subpanels (e.g. a&b and c&d) could be grouped more clearly.

- If two time series are merely shifted versions of each other, a purely analytical approach based on the Fourier transform could align them without the need for a neural network to estimate the phase shift (e.g. by saying that one of the two is the “unshifted prototype” of the other time series). It is hard for me to understand why we need a neural network architecture for this, since if the phase shift between two time series can be determined analytically in the Fourier domain, the guidance network’s task of estimating phi seems redundant.

This connects to the question of how to determine $\phi_0$, since there is an additional symmetry breaking required in the guidance network by randomly designating certain samples as “unshifted” to train the network. Does this introduce potential inconsistencies in the latent space representation across training runs which might learn different mappings for the "unshifted" state, depending on which samples are chosen as the reference? The authors do not explicitly argue that the guidance network is introduced to handle more complex cases with non-uniform shifts or noise, which would make sense to me as a motivating factor. An interesting follow-up question could be what the model actually does if there are non-uniform shifts. How robust is the guidance network’s output for longer/shorter time series?

---

> ### Author Response · Authors · 2024-11-16
>
> We thank the reviewer for recognizing our efforts and for noting that the paper is clearly written.
> We would like to clarify a few of the questions below.
>
> We agree that phase shifting is a well-established technique in signal processing. However, our approach introduces a novel utilization of phase shifting, which has not been proposed in the literature before to the best of our knowledge. We outline the key distinctions in **Appendix G** (Expanded Review of Related Work, page 35 in the original manuscript and page 37 in the revised script). As we shared the same concerns with the reviewer, our section starts by stating “Applying phase shifts to time series is commonly used in signal processing (Haykin & Veen, 2002; Oppenheim et al., 1996)” and lists the key novelties and differences in our approach.
>
> It is important to note that phase shifting alone does not guarantee invariance. Our method achieves this by uniquely encoding each point in the shift space with the phase angle of a specific harmonic, resulting in 100% invariance without the need for a guidance network, as shown in our ablation study.
>
> A simple thought experiment highlights the novelty and distinction of our work. Consider two time series, $\mathbf{x}(t)$ and $\mathbf{x}(t-t^{\prime})$, where the latter is a randomly shifted version of the former, with no information about the shift $t^{\prime}$. How can one map these two samples to a single point in data space without knowing the exact location of $\mathbf{x}(t-t^{\prime})$ and without observing $\mathbf{x}(t)$?
>
> The trivial phase shift does not address this question, nor does the guidance network, hence this is the unique contribution of our paper.
>
>
> We appreciate the reviewer’s suggestion on latent representation visualization. In response, we conducted experiments on electrocardiogram (ECG) signals. We shifted a sample 50 different times and obtained the embeddings. Then, we calculated the Euclidean distance between embeddings with and without applying our proposed transformation. We also illustrated the t-SNE of embeddings. However, as the t-SNE operates on pairwise similarity rather than exact distances, it uses a probabilistic approach that can distort relationships. Therefore, we provided the additional pairwise Euclidean distance between embeddings.
> The results, included on page 36 in the revisited script **Appendix F**, reveal that while shifted samples spread across the latent space and occasionally lead to misclassifications, our approach consistently maps all shifted samples to closer points (the difference is on the order of 1e-6), shrinking the input space significantly.
>
>
> Regarding the comparisons to other approaches in time series alignment, we had a dedicated section for that in **Appendix D.3.** Diffeomorphisms in deep learning on page 30 in the original submission and 31 in the revised submission. For example, one of our comparisons, sequence temporal transformations (STN) from Ohet al. (2018) applied trivial phase shifting with a neural network, however, their method does not provide shift invariance.
>
>
> **Questions**
>
> 1) In the revisited script, we added a couple of additional references for Figure 1. Please let us know if anything can be improved.
>
> 2) We agree that in theory, the phase shift between two time series could be calculated analytically in the Fourier domain. However, our model operates on a single sample at a time during both training and inference, without access to a reference sample for direct comparison. Therefore, we designed the model to learn implicit reference points during training, allowing it to establish alignment without requiring an explicit prototype sample.
>
>
> In our experiments, we evaluated the model across five time series tasks with segment lengths ranging from 200 samples (e.g., PPG) to 4000 samples (e.g., ECG) to demonstrate the model's robustness across varying sequence lengths and types.
>
> We thank the reviewer for the questions and suggestions, which have enhanced the quality of our paper. We welcome any additional experiment requests that would help validate our approach further.

---

> > ### Comment · Reviewer_Se81 · 2024-11-25
> >
> > Thanks for the novel results and the attempts at clarifying our concerns, and apologies for the delayed response. I have to admit I do not have so much familiarity with some of the methods discussed in this paper, so I am trying to balance my own ignorance vs. confusion caused by unclear presentation.
> >
> > But there are some things I remain confused about even after reading the novel Appendix F. You write that: "We agree that in theory, the phase shift between two time series could be calculated analytically in the Fourier domain. However, our model operates on a single sample at a time during both training and inference, without access to a reference sample for direct comparison. Therefore, we designed the model to learn implicit reference points during training, allowing it to establish alignment without requiring an explicit prototype sample."
> >
> > My previous question was about these implicit reference points, which I would interpret as the model introducing a symmetry breaking by designating some (potentially arbitrary) "unshifted" version. Since the mapping is a diffeomorphism, the resulting data presentation are still just time series, so after training you can explicitly see what time series the model designates as shifted vs. unshifted, and whether this assignment of the "implicit reference points" is consisted across runs/there is something to be understood about how it is assigned?

---

> > > ### Author Response · Authors · 2024-11-27
> > >
> > > We thank the reviewer for their feedback and for taking the time to look at the changes.
> > >
> > > Regarding the angle assignments across seeds, we expanded **Appendix F**.
> > >
> > > Specifically, we investigated the behavior of the angle assignments across different random seeds. In the revised script (Appendix F, page 37), we include two new experiments.
> > > The first experiment with our toy example shows that angles may deviate between runs, but the Euclidean distances between shifted samples remain stable. This might suggest that the model achieves a consistent geometric alignment of the data to a degree, even though the specific angle assignments may vary.
> > > Additionally, we repeated the random seed experiment with a real dataset and investigated the assigned angles. Similar to our first experiment, we observed that the assigned angles can vary between runs; however, the Euclidean distances between the samples remain unchanged. This behavior can be interpreted as a system with multiple solutions. In other words, while the assigned angles vary slightly between runs, the Euclidean distances between samples consistently convergence to similar values. We provided these findings in the revised script Appendix F on page 37, including explanations and visualizations.
> > >
> > >
> > > If further clarification is needed, we would be happy to provide additional experiments or explanations.
> > >
> > >
> > > Additionally, we would like to ask if there is anything further we can clarify regarding the differences between our approach and trivial phase-shifting (given in **Appendix G**) and the comparisons between time series alignment tasks (given in **Appendix D.3**) as there were some initial concerns about that. We believe our contribution is distinct from trivial phase-shifting, and we are happy to address any remaining questions or provide additional explanations as requested.
> > >
> > > We thank the reviewer again for their thoughtful feedback and for helping us improve the clarity and depth of our work.

---

> > > > ### Comment · Reviewer_Se81 · 2024-11-28
> > > >
> > > > Thank you for the clarification and additional results in Appendix F. To my mind this actually clears up some things: the phase angles gets randomly assigned by the guidance network at the beginning of training (this was the "symmetry breaking" I was talking about, which as your results demonstrate is naturally achieved by the random seed you assign), and ends up shifting samples in a way to optimize the downstream task.
> > > >
> > > > I agree with Referee iuou that I appreciate the efforts, thoughtfulness and experiments, but I also have to agree that the way the method is introduced and the terminology adds some complication that might still be improved.
> > > >
> > > > I went through the paper again from the beginning and for instance still found the transition between Section 2 and 3 confusing.
> > > > One suggestion could be to make it clearer that while you now have this diffeomorphism in theory, for real-world samples you don't have ground truth phase shift angles, so you need to find them using a differential deep learning framework optimized on some downstream task (e.g. classification, but there might be others). This could also be a place to integrate the ablation studies in Appendix F into the main text.
> > > >
> > > > The same holds for the main contributions that you highlight in Appendix G, which could align more into the contributions on page 2, from which the novelty of you work is not as apparent. In Appendix G you mention that the novel loss term is a crucial novelty aspect of the proposed framework, but this drops somewhat out of the blue on page 5, and is not listed in the main contributions on page 2.
> > > >
> > > > Given the efforts and clarifications, I have now also increased my rating, but in case of acceptance would recommend the authors to integrate some of these suggestions in the camera ready version.

---

> > > > > ### Author Response · Authors · 2024-11-28
> > > > >
> > > > > Thank you for your additional feedback and taking time to read through paper from beginning. We are happy to hear that the additional experiments and results helped clarify your questions.
> > > > >
> > > > >
> > > > > We also greatly appreciate your new suggestions on improving the flow of the manuscript.
> > > > > In response, we have explicitly mentioned about the loss term in the contributions section on page 2 in the revisited script.
> > > > > Additionally, to improve the transition between Sections 2 and 3, we added a new paragraph to the revised manuscript about the need for optimizing the proposed diffeomorphism with a guidance network as the reviewer's suggestion.
> > > > >
> > > > > We also acknowledge the point regarding the terminology and its introduction. We will work to further improve the flow and presentation of the mathematical details to make the explanation clearer.
> > > > >
> > > > > Thank you once again for taking the time to review our paper thoroughly and for appreciating our efforts.
> > > > > Your constructive feedback, especially regarding the additional experiments about the guidance network, helped us improve the paper significantly.

---

### Official Review · Reviewer_Sy9x · 2024-11-02

**Soundness:** 3
**Presentation:** 4
**Contribution:** 3
**Rating:** 8
**Confidence:** 4

**Summary:**

The paper introduces a diffeomorphism that transforms observations from one high-dimensional data manifold to another of the same dimensionality. The goal is to quotient with respect to random shifts. The method ensures that each sample is mapped to a unique point on the data manifold regardless of its random shift. Keeping the same dimension in the representation space is suppose to retain all task-relevant information without any loss.
A theoretical analysis is performed as well as intensive experimental demonstrations.

**Strengths:**

This paper is well motivated and with both theoretical analysis and deep experimental studies and comparisons.
The modeling is interesting and remarks on the performances are relevant.

**Weaknesses:**

The model requires that the data lives in a Manifold. This is a strong assumption (although good one to start with) which sounds necessary as it is quotiented by the group of shifts.

**Questions:**

Data do not necessaraly lives in Manifold, if it does not (no local representation with metrics), would that be an issue?

---

> ### Author Response · Authors · 2024-11-16
>
> We thank the reviewer for this thoughtful question and their support.
>
> For a rigorous mathematical proof, we assume that the data lies on a manifold, as this assumption allows us to work with the time series and shifts in a mathematically precise way. However, our proposed transformation does not strictly require the data to be on a specific manifold. In fact, the transformation would function in basic Euclidean space $\mathbb{R}^n$ as well. What is essential though, is the definition of an input space with local representations and metrics, as these properties support the consistency of our proposed approach in different time series tasks.
>
> Once again, we sincerely thank the reviewer for their comments and for highlighting the significance of our contributions.

---

### Official Review · Reviewer_6oim · 2024-11-05

**Soundness:** 3
**Presentation:** 2
**Contribution:** 3
**Rating:** 6
**Confidence:** 3

**Summary:**

This paper proposes a method to modify time-series data samples in such a way to make neural network training shift-invariant. Their method involves mapping the signal into the frequency domain via the Fourier transform, applying (learned) phase angle transformations to each of the frequency components in the frequency domain, and then mapping back into the time domain via the inverse Fourier transform. They compare results on mostly health-related time-series datasets and compare to alternative methods of promoting shift-invariance such as anti-aliasing methods which use LPFs to remove high-frequency aliasing effects, data augmentation methods, and more. The authors find on average that their method results in improved accuracy and consistency of models on time-series data where shift invariance is desired.

**Strengths:**

The results seem to indicate on average that the models trained in this manner achieve shift invariance (through shift consistency) and better results than existing methods. Theoretical results support that you can achieve a bijective mapping for samples that promotes shift invariance.

**Weaknesses:**

I am admittedly not the most knowledgeable in this area of research, but the method sounds to me like a form of data augmentation that operates in a model-driven manner given that there is a guidance network that is optimized which selects the phase angle to use in the augmentation? Perhaps it would be worthwhile to compare this method to some model-driven / data-driven data augmentation processes?

Some of the results indicate improvement when averaged over multiple runs (how many runs are these results averaged over? I seem to have missed this information in the manuscript). But in some of those cases based on the standard deviations computed there is a fair overlap between prior methods and your method, it’s possible that the advantage in those cases is not so clear when there is a large variance over the accuracy of different runs.

I could imagine there are cases where the computational overhead of training two networks (classifier + guidance network) to get a small increase in accuracy may not be desirable, but as far as I can tell the shift consistency being 100% using this method would seem to be a benefit to any use-case that requires shift invariant predictors, as compared to whichever existing methods are reported in this paper.

I assume that the guidance network, after training, is fixed meaning that the angle for each sample is fixed? How can I interpret the value of the angle that the guidance network converges to for a given sample? Does it say something about the data itself? How robust is this technique to alternative ways of choosing the desired angle, perhaps via random angle methods (could be an interesting baseline to report) or some other method that doesn’t involve optimization of a guidance network?

**Questions:**

If I understand correctly the model is chosen based on the best validation loss, is the reported number based on that validation loss or a held-out test set evaluated by the model whose parameters come from that point where there is the best validation loss during training?

Is it possible that the guidance network is overfitting to the train set in some way? Specifically, if the numbers reported are when you pick the networks that achieve the best validation performance and the reported numbers are not from a held-out test set then there might be some overfitting reported, although the shift consistency metric does seem to indicate that the models are learning shift-invariant estimators.

---

> ### Author Response · Authors · 2024-11-16
>
> We would like to thank the reviewer for the support and insightful questions. We have addressed each point below.
>
> After adding the guidance network to our method, we shared the same concerns with you (Does the improvement only come from feeding different shifts to the model ?) Therefore, we conducted additional experiments. In **Appendix D.3** “Diffeomorphisms in Deep Learning” on page 30 in the original submission and 31 in the revised, we compared our approach with methods that learn transformation functions (data-driven data augmentation). The shift-consistency and performance of our approach significantly outperformed those methods.
>
> Second, in **Appendix D.4** “Expanded Comparison” on page 31 (page 32 in the revised script), we incorporated data augmentation to the previous methods (baselines) and showed that our proposed method still outperforms them in both shift consistency and classification performance.
>
> As a final note, normally, the data augmentation increases the performance and robustness by increasing the diversity of the samples, i.e., AutoAugment. However, in our method, our employed loss functions aim to decrease the variance of angles (less diversity) while considering the overall classification.
>
>
> For evaluation, we ran the experiments three times with different random seeds and averaged the results. This information was given in **Appendix C** in the first paragraph, which we emphasized in the revised script.
>
>
> We agree that on some datasets, performance improvements may seem marginal, especially where baseline models already exhibit saturated performance. However, on challenging datasets such as IEEE SPC, Sleep-EDF, and PhysioNet 2017, our method demonstrates substantial improvements, often reaching 5–10%, while reducing variance across runs.
>
>
> We thank the reviewer for asking about angle interpretation. To provide insight into the angles learned by the guidance network, we conducted a simple experiment. Specifically, we tested the model’s performance in detecting frequencies of two distinct cosine waves (24 and 25 Hz while summed by a 5 Hz sinusoidal). In this case, the model assigned angles to the signals from different classes such that their Euclidean distances were maximized. We believe this can be interpreted as shaping the data space for better separability.
>
>
>
> However, when we conducted the same experiment for larger datasets with more than five classes, interpreting the angle became more complex due to the increased class variability. We have included this new experiment in the revised manuscript on page 36 **Appendix F** to further discuss this point.
>
>
> Regarding the suggestion of random angles, to provide the shift-invariant model, we have to assign random angles to each instance, or some instances should share the same angle during inference, as the optimization of this procedure is expensive. We performed a couple of experiments where all the samples were mapped using a single angle which is determined randomly. However, we have not observed any significant change compared to the initial experiment we performed. We believe that assigning a fixed, single angle can increase the inter-class similarities between samples such that the performance decreases.
>
> ***Questions***
>
> 1) The reported numbers are based on a held-out test set which the model has never seen before. Yes, the evaluation is performed by the model whose parameters come from the best validation loss during training (the test set was never used to save the model).
>
> 2) Although the answer to the first question may explain this, we would like to emphasize again that the reported numbers are from a held-out test set which the model has never observed, we, therefore, do not think there is an overfitting.
>
>
>
> Once again, we sincerely thank the reviewer for their insightful comments and suggestions, as well as for highlighting the angle interpretation, which has contributed to strengthening the quality of our work.

---

> > ### Comment · Reviewer_6oim · 2024-12-02
> >
> > Thank you for your clarification and added details, after reviewing the updated manuscript the paper looks like it is improved. I saw that another reviewer mentioned that it may seem difficult to read the paper due to the "mathiness" of some explanations and I tend to agree. After spending some time with the various expressions used, and cross-referencing with the code sample provided in the appendix, I am reasonably confident I can parse your math but an average reader may find it daunting to understand. If the paper is geared toward practitioners and has the intent of influencing an engineer or research engineer who may benefit from your ideas in some way then I do think the exposition can be streamlined to that effect. For a math-minded reader I don't anticipate any problems with understanding your work. Thanks again for your response and I am confident this work will be accepted given the updated scores by other reviewers as well.

---

> > > ### Author Response · Authors · 2024-12-03
> > >
> > > Thank you for your feedback and support. We are happy to hear that the revisions addressed your questions and improved the paper. We also appreciate that you pointed out that the code sample provided in the appendix helped, we will explicitly refer to this section in the main text as well to make it clearer for future readers.
> > >
> > > Thank you once again for taking the time to review our paper and for appreciating our efforts.

---

### Official Review · Reviewer_iuou · 2024-11-07

**Soundness:** 2
**Presentation:** 1
**Contribution:** 3
**Rating:** 8
**Confidence:** 2

**Summary:**

The paper describes an approach to make neural network pipelines for time series shift invariant. The method is based on Fourier theory and follows a canonicalization approach, where the input signal is shifted by picking a reference frequency and its phase, then shifting the signal such that the phase of that reference frequency equals a target phase φ. The target phase φ is predicted using an invariant network that takes the power spectrum as input. The experiments show strong results across 5 different tasks.

**Strengths:**

* The empirical results are strong and comprehensive across multiple tasks
* The method has clear practical value for making neural networks more reliable for time series analysis
* The problem setting is highly relevant to real-world applications
* The proposed solution addresses an important gap in making neural networks for time series more robust

**Weaknesses:**

1. Clarity and Presentation Issues:
* Despite attempts to make concepts intuitive through figures and explanations, the paper raises more questions than it answers
* The theoretical framework around diffeomorphisms feels disconnected from the practical method
* Several key concepts and proofs need better explanation or justification
2. Missing Context:
* The paper lacks discussion of related canonicalization approaches, particularly work like Kaba et al. (2023) on "Equivariance with learned canonicalization functions"
* The relationship between the theoretical foundations (e.g., Proposition 2.1) and the practical method isn't clearly established

``Kaba, S. O., Mondal, A. K., Zhang, Y., Bengio, Y., & Ravanbakhsh, S. (2023, July). Equivariance with learned canonicalization functions. In International Conference on Machine Learning (pp. 15546-15566). PMLR.```

**Questions:**

This is a mix of comments and questions:

* At the beginning of the notations sections why write parenthesis around $\mathbf{x}$, i.e., why write $(\mathbf{x})$?
* Why is proposition 2.1 relevant. This is a well-known result, but I don't see the paper reflect back on the group structure? In essence, each $e^{j \omega t$ is an irreducible representation of $SO(2)$ and this explains the Fourier shift theorem. The text however, does not really connect prop 2.1. to the rest of the paper.
* I don't understand why fixing $\phi$ would not work. In principle that would make the entire pipeline shift invariant too right?
* The talk about diffeomorphisms is really unclear. What precisely is the diffeomorphism and between which spaces? It seems to be a diffeomorphism of the data-space onto itself, simply by a shift, but I might be misunderstanding something here.
* The last step of the proof of theorem 2.3 is unclear. The first equation of (6) shows that the difference between the two transformed shifted signals is not equal to 0 (but $[T_0/T \omage_0 - \omega]t'$). But then all of a sudden the second equation equates the two signals.
* It is unclear why a variance loss is necessary? More importantly, what is the random variable in this case. It seems like the output of the guidance function is a single angle, then what is there left to take a variance over? The text says "reduce variance in the angles" but the network only returns one angle, right?
* Fig 3 with the depiction of the "manifolds" in (b) is confusing. I still don't understand the manifold viewpoint.
* The text says "In other words, if we conceptualize the dataspace of samples as expanding with shift invariants", what does it mean to expand with shift invariants? And what does it mean "potential points where samples can be found".
* Regarding fully convolutional networks. What precisely is meant with this? I was assuming that this meant that no down-sampling is used, or at least that the output has the same resolution as the input. But then the guidance network does not take a spatial signal as input, but rather the power spectrum, so convolution does not make sense to me, but also even it would, if you return a single phase some sort of pooling or fully connected layer must have been used, right? And thus it cannot be an FCN. Could it be that fully connected is meant here?
* In the section time delay as adversary, what table is being discussed here? Please clarify this in text.
* "However, our results indicate that the model ... the model prediction jumps more than 100%" what does this mean? How can a model "jump" 100%?
* "One distinct results ... increases by 7-8%", relative to what?
* I don't understand why fixing $\phi$ does not work. Theoretically seen, doesn't fixing $\phi$ also lead to a fully invariant pipeline? But in table 5 I see that the first row gives poor performance, why would that be?
* Equation (9) is, similar to my previous question about variance, puzzling to me. Firstly, what is the random variable the variance is taken over, and secondly why would this even work at all? First you minimize variance, now you maximize variance, and the results are barely different. I would expect one of the two to be extremely unstable.
* "Importantly, adding the guidance network does not bring any additional parameters", this is not true. $f_{G_\theta}$ has parameters, right?

---

> ### Author Response · Authors · 2024-11-16
>
> We would like to thank the reviewer for their detailed feedback and for listing their questions one by one, which we have addressed below. We also appreciate their recognition of the practical relevance of the problem and the significance of our proposed solution in real-world applications.
>
>
> From all those questions, fixing $\phi$ and diffeomorphism were emphasized several times. We gave a detailed explanation for each of them.
>
>
> ***Why not fixed $\phi$?***
>
> To provide context, we first refer to prior work. A substantial body of literature has explored the trade-off between accuracy and invariance [1–6], a topic rooted in early learning methods [6, 8].
> For example, [1] reports that increased robustness in models can sometimes reduce standard accuracy, [2] demonstrates that excessive invariance may cause misclassification of some trivial samples. Although these studies focus on adversarial examples, recent work on transformation invariance [3, 4] reveals similar findings.
>
> Furthermore, several studies emphasize that the way the model learns the invariance is critical for performance, showing that invariance does not necessarily correlate with improved accuracy [7].
>
> When we set the $\phi$ to a fixed random value $(-\pi, \pi]$ for all samples, the whole input space shrinks exponentially. Although shrinking the data space can simplify the input and improve the generalization, doing it blindly can bring the inter-class samples closer together in the data space.
> Regarding the suggestion of Reviewer Se81, we designed a simple experiment where a model aims to classify two sinusoidals according to their frequencies. When we observed the assigned angles, we could see that the model assigned angles to increase the Euclidean distance of different classes (frequencies). We refer the reviewer to **Appendix F** on page 36 of the revisited script. Although this result is preliminary with a simple example, it should give an intuition regarding the necessity of understanding data space before shrinking (mapping infinitely many points to a single point) it blindly.
>
>
> [1] Robustness May Be at Odds with Accuracy. ICLR 2019.
>
> [2] Fundamental Tradeoffs between Invariance and Sensitivity to Adversarial Perturbations. ICML 2020.
>
> [3] Fundamental Limits and Tradeoffs in Invariant Representation Learning. JMLR 2022.
>
> [4] Understanding the Generalization Benefit of Model Invariance from a Data Perspective. NeurIPS 2021.
>
> [5] The Double-Edged Sword of Implicit Bias: Generalization vs. Robustness in ReLU Networks NeurIPS 2023.
>
> [6] Learning The Discriminative Power-Invariance Trade-Off. ICCV 2007.
>
> [7] In What Ways Are Deep Neural Networks Invariant and How Should We Measure This? NeurIPS 2022.
>
> [8] Hints and the VC Dimension. Neural Computation. 1993.
>
>
> ***Why Manifolds with Diffeomorphism and the importance of Prop 2.1?***
>
> Our proposed transformation function is a diffeomorphism between input spaces (time series). To provide a rigorous mathematical proof of the bijective proposed transformation, we have to define and operate on these terms. We also cannot take the Fourier transformation of a time series without defining spaces of square-integrable functions $(L(\mathbb{R}^2))$ [9]. In other words, we cannot do calculus without a viewpoint of manifolds.
>
> [9] On convergence and growth of partial sums of Fourier series. Acta Math. 1966
>
> The time shift as a Group operation is common in ML literature. But, not its effects from the harmonic analysis point. The time shift defines separate circle groups for each harmonic depending on the frequency. Since each time series can be decomposed into its harmonics, we observed that the whole shift space can be uniquely defined using a specific harmonic with a period $T_0$; the two paragraphs after Prop 2.1 explain this. To the best of our knowledge, this observation is unique to our method. And, our method cannot be performed by picking a random frequency component as its phase angle will not satisfy the bijective mapping. Therefore, Prop 2.1 is important. For example, if we select the 5 Hz harmonic for a 2-second signal, the mapping of shift space (0-2 seconds) to the phase space (0-200 ms) will not be unique (no injectivity → no diffeomorphism). Shifting the signal with 200ms and 400ms will map to the same phase angle of 5Hz.

---

> > ### Author Response · Authors · 2024-11-16
> >
> > **Questions and the missing context**
> >
> > 1) We do not see any parenthesis around x at the beginning of the notation section. Only the second entry has parenthesis because it represents the Fourier transform.
> > 2) Proof of Theorem 2.3 first shows that all the phases of harmonics are the same after the transformation, also the magnitude is the same so they are equal. If you specify the line number in the proof, i.e. 42-43, we can add a clarification.
> > 3) We provided an answer for variance loss previously. For the loss term, similar to cross-entropy, we take the variance of the angles for a batch of samples. We changed the representation in the revised script.
> > 4) The text says something different. It says “In other words, if we conceptualize the data space of samples as expanding with shift variants, as illustrated in Figure 3, the model learns to reduce the potential points where samples can be found.” It says expanding with shift variants, not invariants. Shift variants of a sample expand the data space. Transformation of an input expands the input space, which is also observed in previous ML papers [4, 10, 11]. Potential points where samples can be found refer to the points where a transformation of a sample can be found.
> >
> > 5) Yes, in the Appendix, it was given correctly but in the main section, it was given as full convolutional, we fixed it to be fully connected in the script, thanks for pointing that out.
> > 6) We explicitly indicated the table in the time delay as an adversary section.
> >
> > 7) The model does not jump, as it is stated the model’s predictions jump by 100%, for example, the heart rate estimation goes from 60 beats per minute to 140.
> > 8) We could not identify the 7-8% line, but if it is 7-10%, it is compared to the baseline. When we applied the previous shift invariance techniques, they decreased the performance in some datasets, i.e., the APS method in DaLiA.
> > 9) We have added a separate section to the revised paper regarding the fixed $\phi$ and increased variance in angles while explaining the preliminary results in our previous answer.
> > 10) Yes, $f_{G_{\theta}}$ has parameters, we gave its details in Appendix C.4.1. The parameter count of $f_{G_{\theta}} $ is approximately 2–4% of that of the main classifier. Our point was that the model capacity for the classification does not increase. Also, we conducted experiments by increasing the number of parameters in the classification model (such as Appendix D.1) to show that the performance increase was not because of the additional parameters.
> >
> > [4] Understanding the Generalization Benefit of Model Invariance from a Data Perspective. NeurIPS 2021.
> >
> > [10] Generalization Error of Invariant Classifiers. AISTATS 2016.
> >
> > [11] Homomorphism Autoencoder—Learning Group Structured Representations from Observed Transitions. ICML 2023.
> >
> >
> >
> >
> > Regarding the related canonicalization approaches, we have included a discussion about the pointed paper in the revisited script (**Appendix G** on page 37), we sincerely thank the reviewer for bringing this paper to our attention. One key point we wish to emphasize is that the mapping in the referred paper is limited to the number of group elements defined by the canonicalization network (an equivariant MLP or G-CNN in the referred paper implementation). Our proposed transformation removes that restriction completely and allows us to map each sample to every point (infinitely many) in the input space without requiring an equivariant neural network. Our method achieves this by uniquely representing each potential shift point as an angle of a specific harmonic of the input—an approach that, to the best of our knowledge, is entirely novel.
> >
> >
> > We sincerely thank the reviewer for this thorough review and their time. Hopefully, these answers have clarified any questions you may have had.

---

> > ### Comment · Reviewer_iuou · 2024-11-25
> >
> > Please forgive my ignorance, but would it be possible to add details on the terminology. E.g., what precisely is meant with *the manifold*. You consider the manifold of square integrable functions, and a single signal x will be referred to as a point in the manifold? Then, the diffeomorphism is a mapping from which manifold to which manifold? (what is the other manifold) And what precisely is the diffeomorphism? If the diffeomorphism is given by T(x,\phi), then is this invertible? It seems like every point in the orbit x(. - t), for every t in the cyclic transformation group, maps to one and the same point (Thm 2.3), but inversely, if we send this canonicalized point back, which point does the inverse land on? If it isn't invertible, then we can't call it a diffeomorphism right? Or pherhaps the diffeomorphism refers to the \Delta \phi prediction? In which case the diffeomorphism is from input signal space to the spaces of tuples that are the orbit representative and delta phi? Like x <-> (T(x,\phi)  , \Delta \phi)
> >
> > The canonicalization method that I referred to before indeed focusses on a finite discrete group, but I was just wondering if the general principle is the same? (predicting a group element from the input x (so g(x)) in an equivariant manner (so g(h x) = h g(x)) such that g(x)^{-1} x is invariant (namely g(h x)^{-1} h x = g(x)^{-1} h^{-1} h x). In your case, your group element predictor is predicting the phase that aligns with that of \omega_0, and use that phase to align the input (canonicalization via the inverse cyclic rotation). The only differences I then see between Kaba et al is that (1) they do this for a finite group whereas you do it for the full SO(2) group (seen as actions on periodic signals), and (2) they do this via a learnable equivariant method whereas you do this via a hand-designed method. Regarding (1), I think this is just a practical considerations as equivariant architectures can be defined for continuous groups too, in the context of the current paper one could e.g. implement such a prediction network via  convolutional neural network implemented in Fourier space. Either way, please do not get me wrong, in no way am I wanting to downplay your contributions, I'm just trying to understand if my viewpoint is correct.
> >
> > I have raised my score to a 5 because I think the paper is sound, however, I still think the core contribution is unecesarily obscured behind terminology and unclear connection to equivariance literature. I also understand I simply may not be fitting the right target audience, but still, I consider myself genuinely interested in the paper.

---

> > > ### Author Response · Authors · 2024-11-27
> > >
> > > Thank you for your questions and for taking the time to provide additional feedback. We sincerely appreciate your interest in the paper and your time.
> > >
> > > We have provided answers to the additional questions below.
> > >
> > > We provided the manifold definitions in **Appendix A.1.2**: Transformations and Diffeomorphism (first page of the Appendix). Also, based on the reviewer’s suggestion, we have explicitly referred to this section in the main section of the revised script.
> > >
> > > We would like to answer some of the questions (the same explanations can be found on the first page of the Appendix). Each observed sample $\mathbf{x}(t)$ is defined to be elements of manifolds $M^{\phi}$ according to the phase angle $\phi(\omega_0)$ of the harmonic with the frequency $\omega_0$. Each signal $\textbf{x}$ represents a point on the manifold, confirming the reviewer's understanding.
> > >
> > > Regarding the question of which manifold to which manifold, we defined samples as elements of manifolds according to their phase angles of harmonic $\omega_0$. For example, as given in the notation list, $M^{\phi_a}$ includes all samples whose harmonic $\omega_0$ has the phase angle of $\phi_a$. Therefore, when we change the phase of the harmonic $\omega_0$ of the sample to $\phi_a$, it is mapped to manifold $M^{\phi_a}$.
> > >
> > > Regarding the invertibility, our transformation $\mathcal{T}(\mathbf{x},\phi)$ with the calculated $\Delta \phi$ is a differentiable bijective function between manifolds which are defined above. Therefore, your understanding of the diffeomorphism from input signal space to the spaces of tuples is correct. We call the whole transformation as diffeomorphism since the $\Delta \phi$ calculation is already performed within $\mathcal{T}(\mathbf{x},\phi)$ and once it is calculated the transformation is a diffeomorphism. And, this was the main reason we included Figure 2 to show the injectivity with $\Delta \phi$.
> > >
> > > Considering the reviewer's suggestion and to make this clearer for the readers, we have changed the transformation output to an explicit tuple in the revised script. Since the proofs already include $\Delta \phi$, we only change the definitions. Also, we have expanded Appendix A.1.2 to provide a detailed explanation of the bijectivity of the transformation function. We also refer the readers to this section of the Appendix in the main script (when we discuss the bijectivity). If the reviewer has further suggestions or concerns regarding the presentation, we are happy to address them and make the necessary revisions. We thank the reviewer for pointing this out.
> > >
> > > Regarding the canonicalization method, we agree with the reviewer that the idea of mapping samples between different data representations is similar. However, we think it is important to highlight that learning a canonical representation involves non-trivial design choices. For example, the authors use shallow networks with a larger filter size to learn finer transformations. And, they mentioned learning finer group representations increases the computation while leading the artifacts. As a best practice, we would like to implement the method and compare it with ours but the design choices with different applications in the original paper made the process slow as we prioritize a fair comparison. That said, we believe our work makes another unique contribution by focusing on temporal signals relevant to human health, such as electrocardiograms. Although we recognize that application-specific contributions are not the central focus of general ML conferences like ICLR, we believe our work addresses an important research gap in addition to having methodological contributions.
> > >
> > >
> > >
> > > We thank the reviewer again for appreciating our efforts and raising the score and the interest in the paper. If there is anything further we can clarify, we are happy to do.

---

> > > > ### Comment · Reviewer_iuou · 2024-11-27
> > > >
> > > > Thank you for the clarifications. I belief the paper is solid and thoughtfully crafted. However, I remain of the position that the technical exposition may be unnecessarily complicated using the diffeomorphism viewpoint, but this should perhaps be treated as a personal opinon based on a "math language barrier" that other readers might also suffer form. Eitherway, I much appreciate the effort spend on clarifying this and improving the readability of the paper, as well providing more context to the method relative to canonicalization methods. An explicit comparison to another canonicalization method would be great to see though.
> > > >
> > > > Given the soundness of the paper, and that the results are good I see no real reason to reject the paper and thus recommend accept (8). I have updated my score accordingly.

---

> > > > > ### Author Response · Authors · 2024-11-27
> > > > >
> > > > > Thank you for your feedback and kind words. We greatly appreciate your acknowledgment regarding the efforts we put into the paper.
> > > > >
> > > > > We will incorporate the quantitative comparison with the referenced canonicalization method, considering both finite and continuous groups, in the manuscript. Your detailed comments and suggestions have been invaluable in strengthening the paper, improving its clarity, and making it more comprehensive. Thank you once again for your time and appreciation of our work.

---

### Meta-Review · Area_Chair_Zayz · 2024-12-15

**Metareview:**

The paper introduces a diffeomorphism-based approach for achieving shift-invariance in deep learning models applied to time series data. The method maps shifted samples to unique points on a high-dimensional manifold without dimensionality reduction, ensuring invariance and preserving task-relevant information. Strengths include a solid theoretical foundation, innovative use of Fourier domain transformations, and empirical validation across diverse time series tasks, demonstrating improved performance and robustness. Weaknesses highlighted by reviewers include the complexity of mathematical exposition, limited clarity in transitions between theoretical and practical contributions, and lack of direct comparisons to other alignment methods. Missing elements include more explicit visualization of learned transformations and further analysis of the guidance network’s performance under out-of-distribution shifts. Despite some presentation issues, the paper’s robust empirical results and theoretical contributions provide strong reasons for acceptance, assuming refinements to clarity and comparisons in the final version.

**Additional Comments On Reviewer Discussion:**

During the rebuttal, reviewers raised concerns about the clarity of the mathematical exposition, the necessity of the guidance network, and the connection between theory and practice. They requested comparisons to other alignment methods, visualizations of latent space consistency, and explanations of the guidance network’s outputs. The authors addressed these by refining the manuscript, adding experiments demonstrating consistent embeddings, providing latent space visualizations, and clarifying theoretical and practical aspects. While some clarity issues remained, reviewers appreciated the method’s robustness, novelty, and strong empirical results, leading to a decision to accept with recommendations for improved presentation in the final version.

---

### Decision · Program_Chairs · 2025-01-22

Accept (Poster)